# α-Hemolysin promotes uropathogenic *E. coli* persistence in bladder epithelial cells *via* abrogating bacteria-harboring lysosome acidification

**Manisha Naskar[1]©, Viraj P. Parekh[2]©, Mathew A. Abraham[3], Zehra Alibasic[1], Min Jung Kim[1], Gyeongseo Suk[1], Joo Hwan Noh[1], Kwan Young Ko[4], Joonha Lee[1], Chungho Kim[1], Hana Yoon[5], Soman N. Abraham[3,6,7], Hae Woong Choi[1]‡ \***

**1** Division of Life Sciences, Korea University, Seoul, South Korea, **2** Department of Biochemistry, Duke University Medical Center, Durham, North Carolina, United States of America, **3** Department of Pathology, Duke University Medical Center, Durham, North Carolina, United States of America, **4** Center for Genomic Medicine, Massachusetts General Hospital, Boston, Massachusetts, United States of America, **5** Department of Urology, Ewha Womans University, College of medicine, Seoul, South Korea, **6** Department of Immunology, Duke University Medical Center, Durham, North Carolina, United States of America, **7** Molecular Genetics & Microbiology, Duke University Medical Center, Durham, United States of America

☙ These authors contributed equally to this work.
‡ Lead contact.
\* haewoongchoi@korea.ac.kr

**Data Availability Statement:** All relevant data are within the manuscript and its Supporting Information files.

## Abstract

There is a growing consensus that a significant proportion of recurrent urinary tract infections are linked to the persistence of uropathogens within the urinary tract and their re-emergence upon the conclusion of antibiotic treatment. Studies in mice and human have revealed that uropathogenic *Escherichia coli* (UPEC) can persist in bladder epithelial cells (BECs) even after the apparent resolution of the infection. Here, we found that, following the entry of UPEC into RAB27b+ fusiform vesicles in BECs, some bacteria escaped into the cytoplasmic compartment *via* a mechanism involving hemolysin A (HlyA). However, these UPEC were immediately recaptured within LC3A/B+ autophagosomes that matured into LAMP1+ autolysosomes. Thereafter, HlyA+ UPEC-containing lysosomes failed to acidify, which is an essential step for bacterial elimination. This lack of acidification was related to the inability of bacteria-harboring compartments to recruit V-ATPase proton pumps, which was attributed to the defragmentation of cytosolic microtubules by HlyA. The persistence of UPEC within LAMP1+ compartments in BECs appears to be directly linked to HlyA. Thus, through intravesicular instillation of microtubule stabilizer, this host defense response can be co-opted to reduce intracellular bacterial burden following UTIs in the bladder potentially preventing recurrence.

## Author summary

Many strains of uropathogenic *E.coli* are capable of secreting the soluble toxin α-hemolysin (HlyA). Intracellular UPEC persistence within bladder epithelial cells is highly

**Funding:** This work was supported by the following grants; the National Research Foundation of Korea grant (NRF-2020R1C1C1003257) to HWC, the US National Institutes of Health grant (K12DK100024) to HWC, the internal grant of Korea University to HWC, and the US National Institutes of Health grants (R01DK121032 and R01DK121969) to SNA. The funders had no role in study design, data collection and analysis, decision to publish, or preparation of the manuscript.

**Competing interests:** The authors have declared that no competing interests exist.

correlated with their ability to express HlyA. HlyA disrupted cytosolic microtubules, which are required for the recruitment of V-ATPase proton pumps to LAMP1$^+$ vesicles. Therefore, the majority of LAMP1$^+$ vesicles carrying UPEC in host cells were unable to be acidified, which is a critical step for killing bacteria. Since HlyA-mediated microtubule fragmentation is essential for UPEC persistence in the bladder, treatment with paclitaxel (a microtubule stabilizer) in the bladders of UPEC-infected mice significantly reduced the bladder bacterial burden even in the absence of antibiotic treatment.

## Introduction

Urinary tract infection (UTI) is the second most common bacterial infection in humans, accounting for approximately 8.1 million clinical visits each year [1]. Women (aged 16–35 years) are approximately 35 times more likely to be UTI patients than men because of anatomical differences, such as a shorter urethra and its proximity to the rectal opening [2]. UTIs are notorious for their capacity for recurrence, even after the administration of an appropriate antibiotic therapy [3,4]. Indeed, recurrent infections typically occur within 6 months after the cessation of antibiotic treatment. It is estimated that up to 25% of UTI patients visiting a clinic have already experienced a previous UTI [5]. The same bacterial strains that initiate the initial infection are responsible for UTI recurrence in patients and can last for up to three years in the urinary tract [3,4].

Uropathogenic *Escherichia coli* (UPEC) accounts for over 80% of UTIs in patients with no underlying pre-existing conditions [5]. UPEC typically initiates infection by attaching to the bladder epithelium, thereby resisting early elimination during voiding. These adherent UPEC rapidly multiply in the urine, exceeding $1 \times 10^3$ colony forming unit (CFU)/mL in this medium [6]. To sustain UTI, a fraction of the urine-borne UPEC gains entry into the normally impervious bladder superficial epithelium by co-opting intracellular RAB27b$^+$ fusiform vesicles, which are distinct membrane-storing vesicles present in these cells to regulate bladder volume [7,8]. These intracellular UPEC can sustain the infection between bouts of bladder voiding, and also promote recurrence of the infection following antibiotic treatment, as in their intracellular location they are protected from antibiotics such as ciprofloxacin, trimethoprim-sulfamethoxazole, and gentamicin that are widely prescribed in clinics [9,10]. The intracellular emplacement of UPEC being protected from antibiotics poses challenges in eradicating UPEC from infected bladders.

Bladder epithelial cells (BECs) possess several distinct mechanisms to reduce the intracellular UPEC load. Almost immediately upon sensing intracellular UPEC within RAB27b$^+$ fusiform vesicles, BECs trigger the expulsion of these intracellular UPEC [7,11]. This involves mobilizing the cellular trafficking machinery employed for hormone secretion to mediate the cAMP-dependent exocytosis of UPEC contained within RAB27b$^+$ fusiform vesicles [8,12]. However, not all the intracellular UPEC are expelled as a recent report has proposed that this is because UPEC escape RAB27b$^+$ compartments via unknown mechanisms to enter the cytosol [12]. This report has suggested that UPEC found free in the cytosol of BECs is promptly recognized by autophagy components, engulfed in autophagosomes, and thereafter expelled in a transient receptor potential cation channel 3 (TRPML3)-dependent manner from lysosomes, once the autophagosomes fuse with lysosomes [12]. UPEC that is not expelled from lysosomes are presumably killed by the bactericidal actions of lysosomes [13,14]. Nevertheless, despite the repertoire of powerful bacteria-clearing actions of BECs, UTIs and their frequent recurrence remain a significant clinical challenge [9,15]. The findings that UPEC form quiescent

intracellular reservoirs (QIRs) within lysosomal vesicles of superficial epithelial cells and the underlying intermediate epithelium of recurrent UTI patients [9] could suggest defects in lysosomal killing within infected BECs.

Many UPEC isolates have been found to secrete soluble toxins, particularly α-hemolysin (HlyA) [16,17] and there appears to be a strong correlation between HlyA expression by UPEC and UTI severity [18,19]. Several clinical studies report that 40 to 58 percent of UPEC isolates exhibit the capacity to secrete hemolysin and that this toxin is directly associated with the onset of UTI [20–22]. Mechanically, HlyA produced by UPEC has been reported to promote cellular toxicity and urothelial damage [16,23]. During UTIs in particular, HlyA was found to promote exfoliation of superficial epithelial cells by triggering caspase-1/caspase-4-dependent inflammatory cell death [24,25]. Interestingly, a clinical study has reported that whereas 37.6% of UPEC isolated from first-time UTI patients express the *hlyA* gene, this number jumped to 48.2% in patients experiencing recurring UTI, suggesting that HlyA might contribute to bacterial persistence in the bladder and recurrent UTIs [26].

Here, we sought to investigate the contribution of HlyA to UPEC persistence in the urinary bladder. We found that HlyA-expressing UPEC promotes intracellular persistence by initially promoting bacterial escape from RAB27b$^+$ vesicles into the cytosol and subsequently preventing the acidification of UPEC-containing lysosomes. HlyA-mediated de-acidification of lysosomes involves the disruption of microtubules involved in trafficking V-ATPase toward lysosomes. Finally, treating mouse bladders with a pharmacological disruptor of microtubules was found to be effective in reducing the intracellular bacterial burden following UTIs.

## Results

### Persistence of UPEC within BECs *in vivo* and *in vitro*

Before investigating the mechanisms employed by UPEC to survive and persist within BECs, we investigated bacterial persistence in the mouse bladder following transurethral UPEC infection. C57BL/6J female mice were trans-urethrally infected with $1 \times 10^8$ CFU of strain CI5, a clinical UPEC isolate, and then the bacterial load in their bladders was assessed at various time-points. On day 1, $2.5 \times 10^3$ CFU of the bacteria were detected in the bladders, but by day 9, the levels had decreased to approximately $5 \times 10^2$ CFU (**Fig 1A**), and after which these residual levels of bacteria remained stable for over 8 weeks (**S1 Fig**). To localize the bacteria within the bladder, we examined cross-sections of the bladder using specific probes for bacteria (red) and the superficial bladder epithelium (green). On day 9, bladder-associated UPEC were found exclusively within the superficial epithelium and co-localized with the lysosomal vesicle marker, LAMP1 (**Fig 1B, arrowheads**). To see if bacterial persistence was attributable to a limited neutrophil response in the bladder, we assessed neutrophil recruitment at various time points following UPEC infection employing flow cytometry, where we probed for Ly6G$^+$ CD11b$^+$ CD45$^+$ cells. As shown in **Fig 1C (**Gating on neutrophils is shown in **S2 Fig**), there appeared to be an early and vigorous neutrophil response within a few hours of infection. However, this neutrophil recruitment began to subside by day 1 and had reached baseline levels by day 6 in spite of the persistence of residual bacteria (**Fig 1C**).

Interestingly, when we subjected biopsied human bladder tissue obtained from recurrent UTI patients to immunofluorescence microscopy, we observed UPEC similarly localized in lysosomal vesicles within bladder epithelial cells (**Fig 1D, arrowheads**; bladder tissue images from additional patients are provided in **S3 Fig**). These observations confirm these findings in mice showing UPEC intracellular localization within bladder epithelial cells of mice is comparable to that in humans.

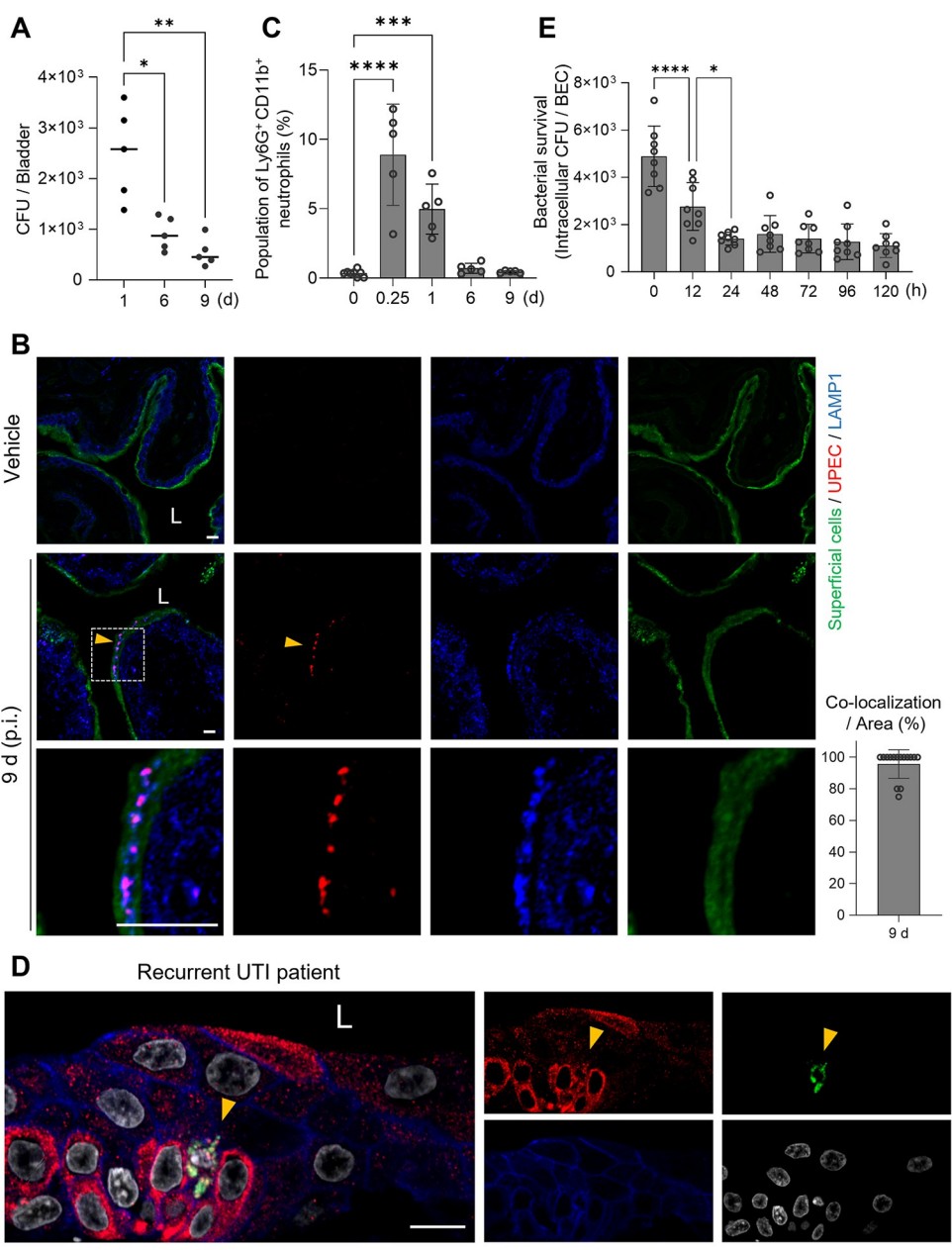

**Fig 1. UPEC persist in bladder epithelial cells. A.** Prolonged bacterial burden in the infected mouse bladders. C57BL/6 female mice were infected by intravesical instillation of the UPEC CI5 strain; bacterial CFUs in the infected bladders were measured at the indicated time-points. Single dot indicates an individual mouse, and the panel represents one representative result from three independent experiments. **B.** Presence of UPEC in infected mouse bladders. Confocal microscopy images of mouse bladders infected with the CI5 UPEC strain for 9 d. UPEC (*E. coli*, red), lysosomal vesicles (LAMP1, blue), and superficial epithelial cells (wheat-germ agglutinin, green). The regions highlighted by squares with dashes in the upper panels are enlarged in the corresponding lower panels. "L": lumen of the bladder. (Bar graph) The percentage of co-localization between UPEC and LAMP1$^+$ vesicles in an 80 μm × 80 μm microscopic region. **C.** Neutrophil response peaked at 6 h p.i. and then diminished. Recruitment of neutrophils was measured by analyzing the flow cytometry in the infected mouse bladders. C57BL6/J mice were infected with UPEC CI5 strain, and then harvested bladders at each time point were analyzed for flow cytometry analysis. Ly6G$^+$ CD11b$^+$ CD45$^+$ cells in the infected mouse bladders were counted by flow cytometry. The population percentage was determined using CD45$^+$ cells. Each dot represents an individual mouse and the panel shows the combined results from two independent experiments. **D.** Bladder biopsies were obtained from recurrent UTI patients and immunostained for UPEC (*E. coli*, green), urothelium (E-cadherin, blue), and lysosomal vesicles (LAMP1, red). "L": lumen of the bladder. **E.** UPEC C15 strain persisted in the infected human 5637 bladder epithelial cells (BECs) for over

120 h p.i. Data information: Quantitative data from two to three independent experiments were analyzed. Data are shown as mean ± SD. Data were analyzed by Kruskal-Wallis test or ordinary one-way ANOVA, *P<0.05; **P<0.01; ***P<0.001; ****P<0.0001. Scale bar: 10 μm.

Having observed that UPEC can persist *in vivo* for prolonged periods within superficial BECs, we sought to elucidate the underlying basis for this phenomenon, by investigating bacterial persistence *in vitro* in the widely employed human 5637 bladder epithelium cell line, which was established from a transitional cell carcinoma of the urinary bladder [27,28]. We found that following exposure of the 5637 BECs to the CI5 strain [$2 \times 10^7$ CFU; multiplicity of infection (MOI) of 100] and subsequent exposure to gentamicin (40 μg/mL), to kill extracellular bacteria, close to $5 \times 10^3$ CFU of bacteria entered the BECs. Thereafter, the numbers of intracellular bacteria steadily declined, as observed *in vivo*, for the next 24 h (**Fig 1E**). However, the numbers of intracellular bacteria within BECs steadied at approximately $1 \times 10^3$ CFU until at least day 5 post infection (p.i.), at which point we concluded these studies (**Fig 1E**). Thus, the ability of UPEC CI5 to persist within the human BEC line *in vitro* appeared to mimic their ability to persist within primary superficial bladder epithelium in mice and in primary bladder epithelial cells in humans.

## Trafficking of intracellular UPEC from RAB27b vesicles into LC3A/B⁺ compartments is preceded by a brief cytosolic phase

Previous studies have revealed that infected BECs employ distinct components of the cellular export machinery, including RAB27b⁺ fusiform vesicles and lysosomal vesicles, to lower the intracellular bacterial burden [8,12,29]. We sought to confirm previous findings that UPEC harbored in RAB27b compartments eventually became encapsulated within LC3A/B⁺ compartments (autophagosomes). We probed UPEC-infected BECs with antibodies against RAB27b and LC3A/B⁺ compartments, at different time-points following infection. **Fig 2A** is a microscopic image taken at 4 h p.i. showing a BEC harboring UPEC closely associated with a RAB27b⁺ vesicle. We found that whereas a majority (70%) of UPEC were housed in RAB27b⁺ compartments at 2.5 h p.i., their numbers dropped to 30% by 12 h and remained at that level even at 24 h p.i. (**Fig 2B**). Conversely, by 5 h p.i., only a small proportion (<30%) of the intracellular UPEC were housed in LC3A/B⁺ compartments, but by 12 and 24 h, the proportion had increased to 70% (**Figs 2C and S4**). Shown in **S4A Fig** is a BEC housing UPEC within an LC3A/B⁺ compartment at 8 h p.i. Moreover, we observed that the UPECs in LC3 autophagosomes at 8 h p.i. did not co-localize with RAB27b molecules, which were present in the vesicles at 4 h p.i. (**S4B** and **S4C Fig**). These findings are consistent with the notion that UPEC are initially encased in RAB27b⁺ compartments but, at later time points, become ensnared in LC3A/B⁺ compartments. To support these microscopic studies, we undertook an independent experimental approach where we sought to isolate intracellular UPEC from infected BECs at various time points p.i. and identify the specific intracellular vesicle they were associated with. To facilitate the isolation of intracellular UPEC, we first crosslinked these bacteria with magnetic microbeads and then exposed them to BECs. Although this cross-linking technique had previously been shown not to significantly impact bacteria's ability to bind and invade BECs [12], we confirmed this by showing that magnetic bead-bound UPEC readily bound BECs as unlabeled UPEC and were internalized to the same degree (**S5** and **S6 Figs**). To isolate vesicle-associated intracellular UPEC, we selectively disrupted the plasma membrane of infected BECs at various time intervals and then employed a magnet to separate the bacteria from the rest of the cellular lysate as described previously [12]. By using this technique followed by subjecting the bacterial isolates to western blot, we confirmed that most of the UPEC were associated with

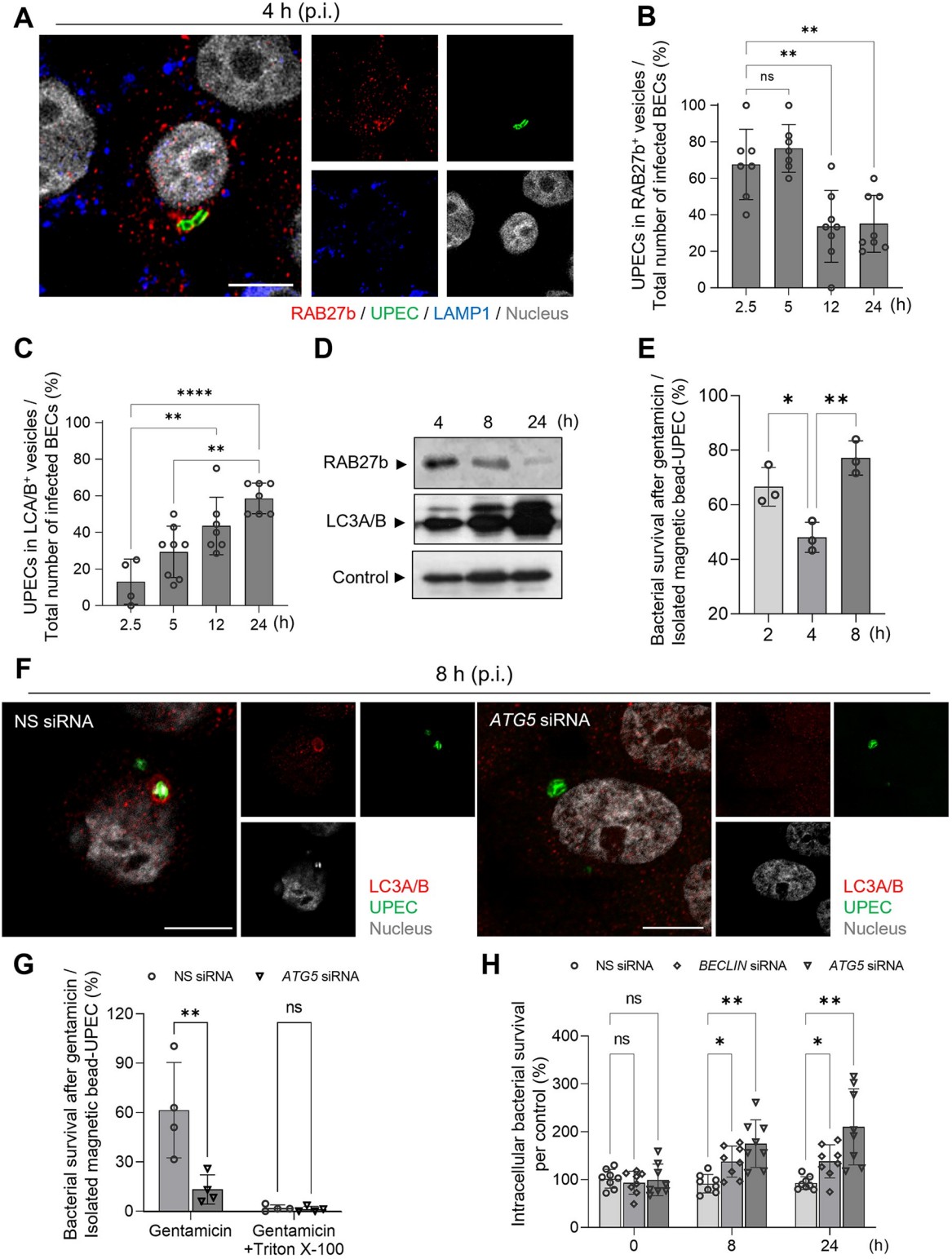

**Fig 2. Autophagosome formation after the escape of UPEC from RAB27b⁺ vesicles. A.** UPEC present in RAB27b⁺ vesicles at 4 h p.i. Human 5637 BECs were infected with the UPEC CI5 strain for 4 h and were stained with anti-RAB27b (red), anti-UPEC (green), or anti-LAMP1 (blue) antibodies. Representative images are from randomly selected regions. **B** and **C.** Reciprocal trend of UPEC presence in RAB27b⁺ vesicles and LC3A/B⁺ vesicles of human 5637 BECs. To quantify the co-localization of UPEC with **(B)** RAB27b⁺ or **(C)** LC3A/B⁺, the optical fields (100 μm × 100 μm) were randomly selected in the confocal microscopy images and quantified at the indicated time-

points. **D.** Prevalence of UPEC in different vesicles during infection. The human 5637 BECs were infected with magnetic bead-labeled UPEC, and the UPEC-housing vesicles were extracted at the indicated time-points. Anti-RAB27b or anti-LC3A/B antibodies were used for western blotting. Loading control of UPEC-containing vesicles was detected with anti-GAPDH antibody. **E.** Time course survival of isolated UPEC from gentamicin treatment. Human 5637 BECs were infected with magnetic bead-labeled UPEC. After the indicated time points, magnetic bead-labeled UPEC were isolated from infected host cells and exposed to gentamicin. No gentamicin treatment on isolated UPEC was used to calculate intracellular UPEC inside or outside of vesicles. **F.** Presence of UPEC in vesicles expressing LC3A/B. The human 5637 BECs were transfected with non-specific (NS) siRNA or *ATG5* siRNA. Then, the UPEC CI5 strain infected human BECs (8 h) were stained with anti-*E. coli* (green) and anti-LC3A/B (red) antibodies. **G.** UPEC in autophagosomes were protected from gentamicin exposure. The human 5637 BECs were pre-treated with NS siRNA or *ATG5* siRNA, followed by infection with the magnetic bead-labeled UPEC CI5 strain. After 4 h of infection, the isolated UPEC containing vesicles were treated with gentamicin and bacterial survival was examined. Gentamicin incubation was performed with or without 0.1% Triton X100. Data was normalized by dividing the "CFU of UPEC in each condition" with the "CFU of isolated magnetic bead-UPEC without gentamicin treatment". **H.** Autophagosome-mediated UPEC trafficking requires BECLIN and ATG5. Human 5637 BECs were transfected with *BECLIN* siRNA (square), *ATG5* siRNA (triangle), or NS siRNA (circle). Transfected cells were infected with UPEC CI5 strains at the indicated time-points, and the intracellular bacterial burdens were compared. Then, the percentage of each condition's CFU relative to the CFU of WT BECs infected with UPEC was calculated. Data information: Quantitative data from two to three independent experiments were analyzed. Data are shown as mean ± SD. Data were analyzed by ordinary one-way ANOVA (B, C, E) or two-way ANOVA (G, H), *P<0.05; **P<0.01; ****P<0.0001; n.s, not significant. Scale bar: 10 μm.

RAB27b$^+$ vesicles at 4 h p.i. However, by 8 and 24 h p.i., the association of intracellular UPEC with RAB27b became markedly decreased (**Fig 2D**). In contrast, low levels of intracellular UPEC were associated with LC3A/B vesicles at early time points, but at later time points the association level markedly increased (**Fig 2D**).

It has previously been proposed that prior to encapsulation of UPEC in LC3A/B$^+$ autophagosomes, the bacteria are likely found free in the cytosol [12]. To demonstrate the existence of cytosolic UPEC in infected BECs, we infected BECs with magnetic beads labeled UPEC. Thereafter, at various time intervals, we selectively lysed the plasma membrane of infected BECs and isolated intracellular UPEC using a magnet and then exposed them to gentamicin. If isolated UPEC were truly cytosolic and not encased in a vesicle, they would display sensitivity to gentamicin (**S7 Fig**). We found that whereas the majority of UPEC isolated at 2 h (67%) and 8 h p.i. (77%) were insensitive to gentamicin, only 48% of UPEC at 4 h p.i. were protected from gentamicin (**Figs 2E** and **S8**). Interestingly, 4 h p.i. corresponds to the timepoint at which UPEC found in RAB27b transition into LC3A/B vesicles. To further support the notion that a cytosolic phase exists between the two proposed vesicular phases, we sought to knock down autophagy-related 5 (ATG 5) which works closely with LC3A/B in forming the autophagosome in BECs [30,31]. We reasoned that if we knocked down a critical component of the autophagosome such as ATG5 (**S9 Fig**), we would prevent uptake of cytosolic UPEC by LC3A/B vesicles in infected BECs (**Fig 2F**). As a consequence, many more intracellular UPEC would be susceptible to gentamicin treatment in *ATG5* siRNA transfected BECs compared to infected non-specific (NS) siRNA transfected BECs. Indeed only 10% of UPEC from *ATG5*-knocked down BECs was protected from gentamicin, whereas over 60% of UPEC isolated from control BECs was protected from gentamicin (**Figs 2G** and **S10**) at 8 h p.i. This was further confirmed because when isolated UPEC from NS or *ATG5*-knocked down BECs were treated with Triton X-100, which disrupts host membranes [12], both groups of UPEC were susceptible to gentamicin.

Finally, we examined whether the cytosolic existence of UPEC is advantageous to bacteria by quantifying intracellular UPEC numbers in WT BECs and in BECs where particular autophagy components were knocked down. We compared intracellular UPEC numbers obtained at different times p.i. in BECs that had earlier been transfected with NS siRNA (control), *ATG5* siRNA, or *BECLIN* siRNA (ortholog of yeast Atg6, another component of the autophagosome) [31]. **Fig 2H** reveals that, compared to control transfected BECs, BECs knocked down in either ATG5 or BECLIN harbored significantly more intracellular bacteria than

control transfected BECs. Further, this number seemed to increase significantly with increased incubation time suggesting that when not restricted in a vesicle UPEC have the capacity to grow rapidly in the cytosol of BECs (**Figs 2H** and **S11**). Therefore, our findings suggest that intracellular UPEC numbers are kept controlled as long as bacteria are encased in various vesicles, but if the bacteria are able to escape into the cytosol, the BECs do not have the capacity to control their numbers.

## HlyA facilitates the escape of UPEC from RAB27b⁺ vesicles

HlyA is a virulence determinant found in isolates that are frequently reported in UTIs and known to cause hemolysis of red blood cells through the disruption of cell membranes. In view of its membrane disrupting properties, we hypothesized that the contribution of HlyA to UPEC virulence in the bladder promotes bacterial escape from the RAB27b compartments. The strain CFT073 is a commonly used HlyA-expressing UPEC, while *ΔhlyD* is a HlyA⁻ isogenic mutant strain [32,33]. We compared the hemolytic capacity of strain CI5 or strain J96 with that of the CFT073 and *ΔhlyD* strains. *E. coli* K-12, which is deficient in HlyA or other hemolytic factors, was also included in this assay. UPEC strain CI5, J96, and CFT073 caused significant hemolysis, with CFT073 causing twice as much hemolysis as CI5, as compared to that caused by the K-12 strain, whereas strain *ΔhlyD* exhibited limited hemolysis (**Fig 3A**). In view of these findings, we proceeded to employ strains CFT073 and *ΔhlyD* to study the contribution of HlyA in bacterial escape from the RAB27b⁺ compartment into the cytosol. The exposure of BECs to the wild-type (WT) CFT073 strain (MOI: 200) resulted in significant cytotoxicity (**Fig 3B**). Therefore, we examined lower MOIs to identify MOIs with limited cellular cytotoxicity. At MOIs of 50 and below, BECs retained their viability for at least 16 h (**Fig 3C**), and we used a sublytic MOI of 50 or below for the CFT073 strain for the remainder of our study.

At this MOI, a comparable numbers of strains CFT073 and *ΔhlyD* gained entry into BECs by 2 h p.i. (**Fig 3D**). To localize the compartments in which they were harbored, WT CFT073 and *ΔhlyD* mutant strains were first covalently linked to magnetic beads and treated to BECs at a MOI of 20 for 2 h. Thereafter, the plasma membrane of BECs was selectively lysed, and intracellular bacteria were harvested using a magnet. These bacteria and associated host membranes were subjected to western blotting using a RAB27b-specific antibody as a probe. We found that both CFT073 and *ΔhlyD* strains were associated with comparable amounts of RAB27b membranes, indicating that at this time-point, both bacteria were still harbored in the RAB27b compartments (**Fig 3E, top panel**). However, at 5 h p.i., although the *ΔhlyD* strain was still associated with RAB27b, relatively small amounts of RAB27b were still associated with the CFT073 strain (**Fig 3E, bottom panel**). Complementation with a plasmid encoding *phlyD* decreased the association of the *ΔhlyD*(*phlyD*) strain with RAB27b (**S12 Fig**). These results indicate that WT CFT073 UPEC could escape these vesicles and enter the cytosol. To determine if escape from the RAB27b compartment was beneficial to UPEC, we compared intracellular bacterial numbers in BECs, at 4 h p.i. and found that intracellular numbers of the CFT073 strain were significantly higher than those of the *ΔhlyD* strain (**Figs 3F** and **S13**). When we investigated the count of extracellular bacteria in BECs infected with the CFT073 and *ΔhlyD* strains and found that the numbers of extracellular *ΔhlyD* strain were significantly higher than those of the CFT073 strain (**Figs 3G** and **S14**). To confirm that the CFT073 strain is indeed cytosolic, we isolated magnetic bead-coated CFT073 and *ΔhlyD* strains at 4 h p.i., after they had been exposed to BECs, and performed a gentamicin-protection assay, as described above. The CFT073 strain was susceptible to gentamicin, whereas the *ΔhlyD* strain was mostly resistant (**Figs 3H** and **S15**). The resistance of the *ΔhlyD* strain to gentamicin is

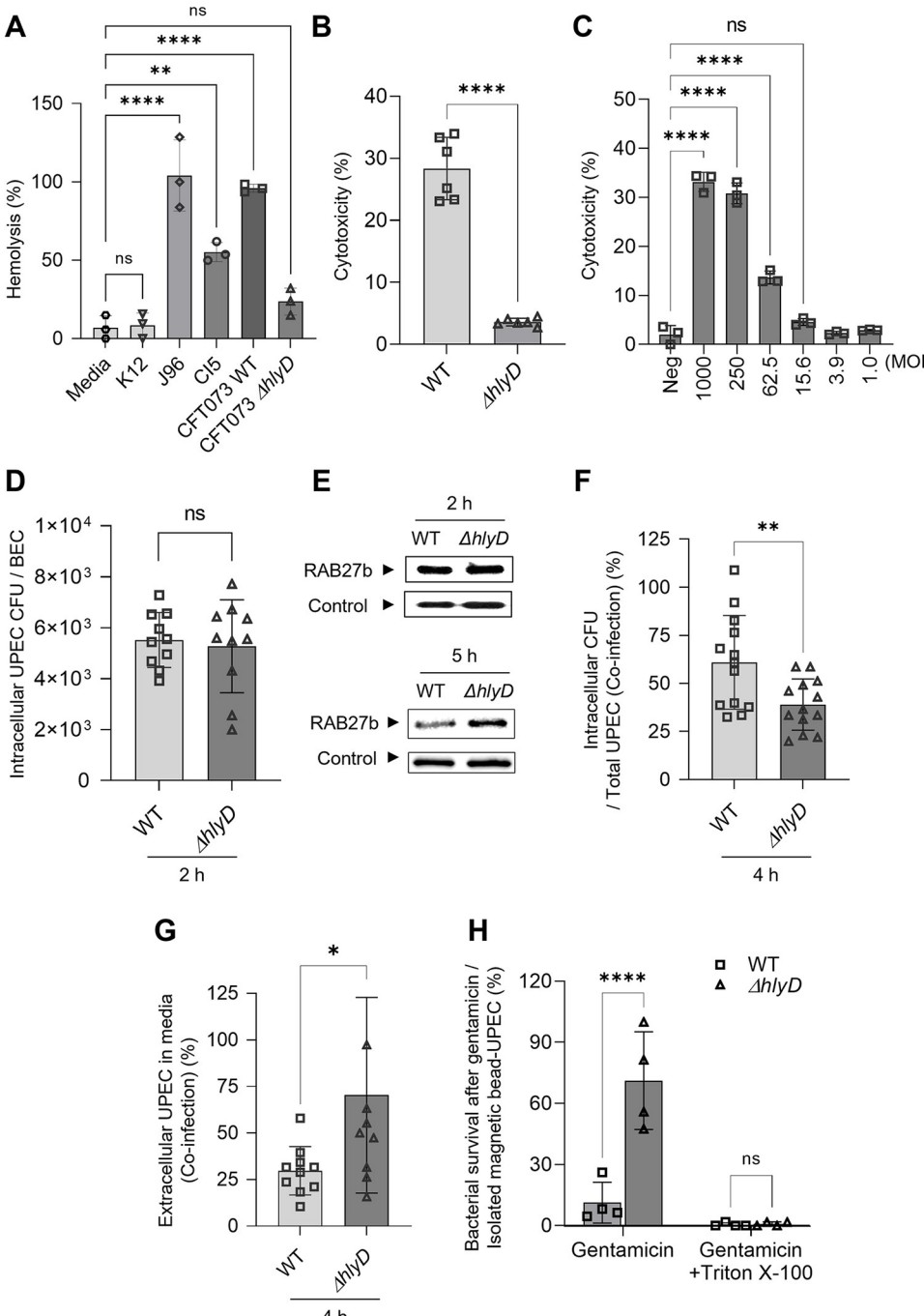

**Fig 3. HlyA-mediated evasion of UPEC from RAB27b⁺ fusiform vesicles. A.** The ability of UPEC strains to mediate red blood cell (RBC) lysis varies. The levels of RBC membrane lysis by various UPEC strains (K12, J96, CI5, WT CFT073, or *ΔhlyD* CFT073) were compared. **B.** HlyA is responsible for BEC cytotoxicity. The human 5637 BECs were infected with the WT or *ΔhlyD* CFT073 strains (MOI: 250). After 16 h, the levels of lactate dehydrogenase (LDH) release were assessed to examine cytotoxicity. **C.** Determination of sub-lethal MOI for the UPEC CFT073 strain. The human BECs were infected with the WT CFT073 strain, in an MOI-dependent manner. After 16 h, the levels of cytotoxicity were assessed by measuring the levels of LDH release. **D.** Equal initial burden of UPEC strains. The human 5637 BECs were infected with the WT or *ΔhlyD* CFT073 strain (MOI: 20), and the intracellular CFU were measured at the 2 h p.i. **E.** HlyA-mediated escape of UPEC from RAB27b⁺ vesicles. The human 5637 BECs were infected with magnetic beads labeled with WT or *ΔhlyD* CFT073 UPEC strain (MOI: 20). At 2 or 5 h p.i., the UPEC-housing vesicles were extracted and immunoblotted using an anti-RAB27b antibody. Loading control of UPEC-containing vesicles was detected with anti-GAPDH antibody. **F** and **G.** Enhanced intracellular bacterial burden with less extracellular

expulsion was mediated by HlyA. The human 5637 BECs were infected with a 1:1 mixture of WT and *ΔhlyD* CFT073 strains, following which the **(F)** intracellular or **(G)** extracellular CFU were measured, at 4 h p.i. **H.** Escape of UPEC from RAB27b⁺ vesicles was mediated by HlyA. The human 5637 BECs were infected with magnetic bead-labeled UPEC WT or *ΔhlyD* CFT073 strain. After 5 h of infection, the isolated UPEC containing vesicles were treated with gentamicin and bacterial survival was examined. Gentamicin incubation was performed with or without 0.1% Triton X100. Data was normalized by dividing the "CFU of UPEC in each condition" with the "CFU of isolated magnetic bead-UPEC without gentamicin treatment". Data information: Quantitative data from two to three independent experiments were analyzed. Data are shown as mean ± SD. Data were analyzed by ordinary one-way ANOVA (A, C), two-way ANOVA (H), or unpaired two-tailed Student's *t*-test (B, D, F, G). *P<0.05; **P<0.01; ****P<0.0001; n.s, not significant.

attributable to its membrane-bound status. When both bacteria were exposed to gentamicin after exposure to Triton X-100 treatment, they were highly susceptible to gentamicin (**Figs 3H and S15**). Taken together, our findings imply that HlyA contributes to the virulence of the CFT073 strain by promoting the escape of bacteria from RAB27b⁺ vesicles into the cytosol, prior to being expelled from the cell by the autonomous defenses of BECs.

## Prolonged persistence of UPEC in BECs is aided by HlyA

Although HlyA promotes the early escape of UPEC from RAB27b⁺ vesicles, this phenomenon is short-lived, as most cytosolic bacteria are recognized by the cell's autonomous defense system and recaptured in LC3A/B autophagosomes (**Fig 2F**). Additionally, these bacteria are once again encapsulated and subsequently shuttled into LAMP1⁺ lysosomes [12,34]. Thus, to investigate where intracellular UPEC were housed in BECs 24 h p.i., we probed UPEC-infected BECs with antibodies against LAMP1, at different time-points following infection. Over 80% of UPEC were in the LAMP1⁺ compartment (**Fig 4A and 4B**), which was supported by the fact that when the magnetically labeled-UPEC were isolated from BECs at 4 and 24 h p.i. and subjected to western blotting, only a few LAMP1⁺ membranes with intracellular UPEC were observed at 4 h, but these levels significantly increased after 24 h (**Fig 4C**). Interestingly, when the incubation time was increased to 72 h p.i., little change was observed in the amount of LAMP1⁺ associated with intracellular UPEC (**Fig 4D**). This result indicated that intracellular bacterial populations had stabilized and that bacteria had developed a long-term survival capacity within LAMP1⁺ lysosomes, which are normally bactericidal and degradative intracellular compartments.

Due to the fact that lysosomes are enriched in proteases and lipases, which are highly bactericidal under acidic conditions (pH 4.5–5.5), we examined the pH of lysosomal compartments containing CFT073 UPEC by applying an acidotropic dye to infected BECs harboring the WT UPEC CFT073 strain, at 24 h p.i. We found that very few of the intracellular WT UPEC housed in LAMP1⁺ compartments stained positive with the dye (**Fig 4E and 4F**), implying that the survival of WT UPEC within lysosomes was due to their ability to prevent acidification of these compartments. Interestingly, when we examined whether the LAMP1⁺ compartment harboring the relatively small population of *ΔhlyD*, harbored in this compartment, became acidic following infection with BECs, the majority of vesicles took up the acidotropic dye (**Fig 4E, white arrowheads, and 4F**). However, complementation with a plasmid encoding *phlyD* led to suppression of acidification of LAMP1⁺ compartments of the *ΔhlyD(phlyD)* strain (**S16 Fig**). Therefore, HlyA assists the bacteria in abrogating lysosomal acidification. Additionally, the number of LAMP1⁺ vesicles harboring the *ΔhlyD* mutant that co-localized with the acidotropic dye was significantly higher than the number of WT UPEC that co-localized with the acidotropic dye-labeled LAMP1⁺ vesicles (**Fig 4F**). These findings suggested that the WT CFT073 strain avoids acidification of the LAMP1⁺ compartments *via* HlyA expression.

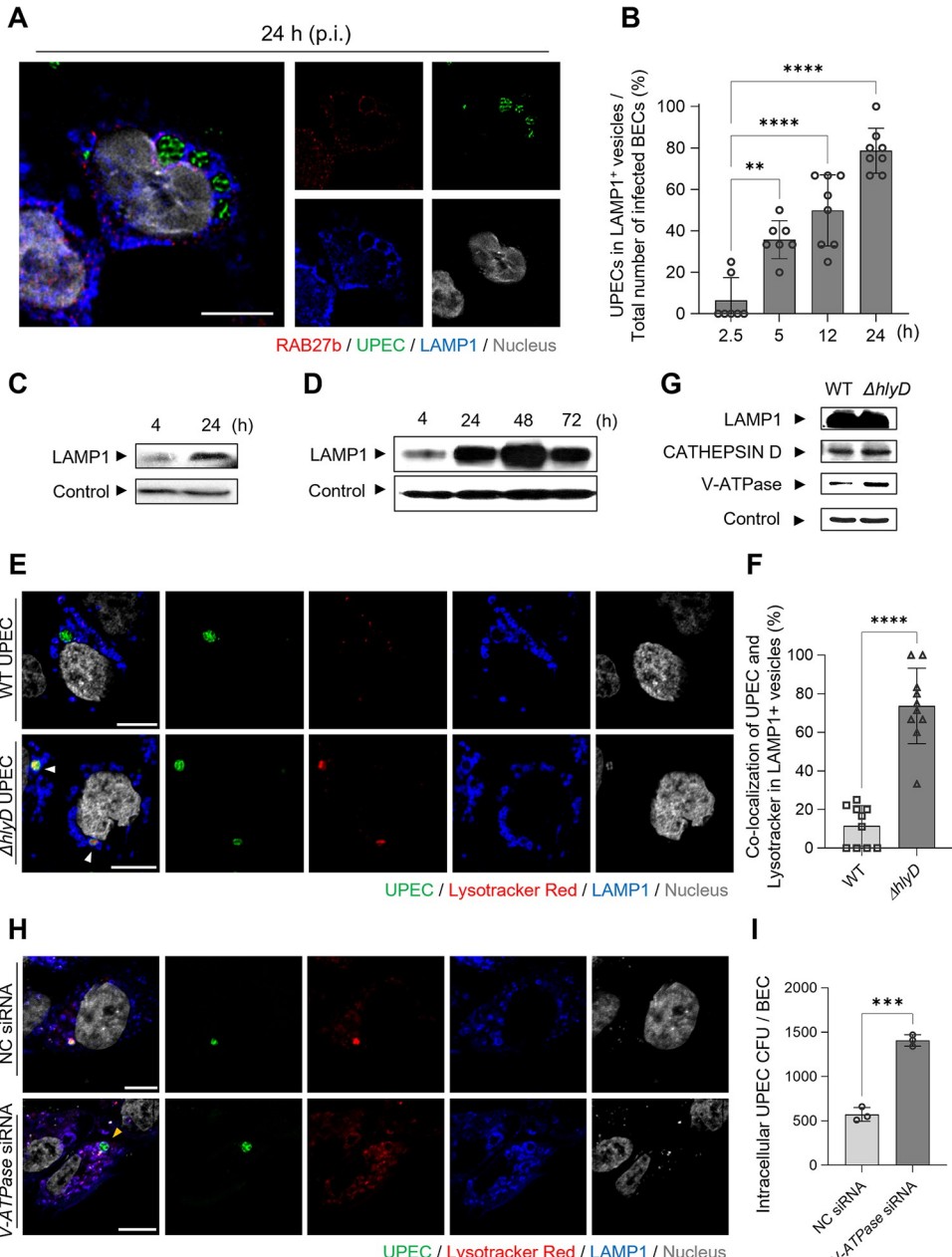

**Fig 4. UPEC persists in LAMP1$^+$ vesicles by preventing the acidification of LAMP1$^+$ vesicles. A.** UPEC lingers in LAMP1$^+$ vesicles during a prolonged BEC infection. After 24 h of UPEC CI5 infection, the human 5637 BECs were stained with anti-RAB27b (red), anti-UPEC (green), or anti-LAMP1 (blue) antibodies. Representative images are from randomly selected regions. **B.** UPEC prevalence in LAMP1$^+$ vesicles gradually increased. To quantify the co-localization of UPEC with LAMP1$^+$, the optical fields (100 μm × 100 μm) were randomly selected in the confocal microscopy images and quantified at the indicated time-points. **C and D.** UPEC can persist in LAMP1$^+$ vesicles for extended periods of time. The human 5637 BECs were infected with magnetic bead-labeled UPEC, and then isolated UPEC-containing vesicles were immunoblotted using anti-LAMP1 antibody. Loading control of UPEC-containing vesicles was detected with anti-GAPDH antibody. **E.** HlyA blocked the acidification of UPEC-containing vesicles. After infecting human 5637 BECs with WT CFT073 or *ΔhlyD* strains for 24 h, LysoTracker Red DND-99 dye was applied and then stained with anti-LAMP1 (blue) or anti-UPEC (green) antibodies for confocal microscopy imaging. **F.** The number of UPEC co-localized with LysoTracke Red dye was quantified in randomly chosen fields of Fig 4E. **G.** HlyA prevented the recruitment of V-ATPase onto the LAMP1$^+$ vesicles. The human 5637 BECs were infected with magnetic bead-labeled WT or *ΔhlyD* CFT073 strain for 24 h. The isolated UPEC-containing vesicles were immunoblotted using antibodies targeting LAMP1, Cathepsin D, V-ATPase, or GAPDH. **H.** Acidification of UPEC-containing LAMP1$^+$ vesicles by V-ATPase. The human 5637 BECs transfected with NS siRNA or V-ATPase siRNA

were infected with *ΔhlyD* strain, and then dyed with LysoTracker after 24 h (red). Anti-UPEC (green) and anti-LAMP1 (blue) antibodies were used to stain the fixed cells. **I.** V-ATPase is critical for UPEC persistence. The human 5637 BECs were transfected with NS siRNA or *V-ATPase* siRNA, infected with the UPEC *ΔhlyD* CFT073 strain for 24 h followed by assessment of bacterial burden. Data information: Quantitative data from two to three independent experiments were analyzed. Data are shown as mean ± SD. Data were analyzed by ordinary one-way ANOVA (B), or unpaired two-tailed Student's *t*-test (F, I). **P<0.01; ***P<0.001; ****P<0.0001. Scale bar: 10 μm.

## Inhibition of V-ATPase recruitment by HlyA

LAMP1$^+$ lysosomes become acidic when V-ATPase associates with vesicle membranes to pump protons into the vesicle lumen in an ATP-dependent manner [35,36]. Therefore, we performed western blotting to determine the presence of V-ATPase in WT and *ΔhlyD* UPEC-harboring lysosomes. The *ΔhlyD* mutant was associated with high amounts of V-ATPase, whereas only minimal amounts of this enzyme were linked to WT UPEC-associated lysosomes, which could explain the lack of acidification in the compartments (**Fig 4G**). We also analyzed the content of the lysosomal content marker Cathepsin D in each compartment, to show that there was limited loss of lysosomal integrity, in UPEC-harboring lysosomes. The amount of Cathepsin D in WT UPEC-harboring lysosomes showed limited difference from that of the *ΔhlyD* mutant-harboring lysosomes (**Fig 4G**). To further assess whether the loss of V-ATPase in host BECs leads to the loss of LAMP1$^+$ acidification, we examined the acidification of lysosomal compartments containing *ΔhlyD* mutant within BECs transfected with NS siRNA or *V-ATPase* siRNA. Confocal microscopy images showed that *V-ATPase* siRNA-transfected BECs failed to contain the *ΔhlyD* UPEC in acidic LAMP1$^+$ vesicles (**Fig 4H**, yellow arrowhead, and **S17**). Pharmacological inhibition of V-ATPase with Bafilomycin A1 also led to a loss of acidification in UPEC-containing LAMP1$^+$ vesicles (**S18 Fig**). Since the loss of acidification of bacteria-containing LAMP1$^+$ vesicles is advantageous for bacterial survival (**Fig 4I**), HlyA potentially plays a critical role in promoting intracellular UPEC persistence in LAMP1$^+$ lysosomes of BECs by preventing the V-ATPase-dependent acidification of lysosomes.

## Destabilization of the cytoskeletal structure in bacteria-infected BECs reduces intracellular bacterial killing

Next, we investigated how a toxin that permeabilizes the lipid bilayer can prevent lysosomal acidification. The cytoskeletal structure of cells compartmentalizes intracellular membrane-bound vesicles, such as lysosomes and autophagosomes, with microtubules playing a critical role in controlling vesicle movement, which can result in autophagosome maturation [37]. Autophagosomes formed in the cell periphery move centripetally along microtubule tracks via dynein to fuse with the V-ATPase-bearing lysosome in the juxtanuclear region of the cell, thereby facilitating autophagosome-lysosome fusion [38,39]. We hypothesized that the lack of acidification in LAMP1$^+$ WT UPEC-containing vesicles could be related to defects in cytoskeleton-mediated trafficking of V-ATPase-containing lysosomal vesicles. When we examined the cytoskeletal structure of saline-treated control BECs and bacteria-exposed BECs, we observed that the microtubules were rearranged and less organized in the bacteria-exposed BECs than in the treated control BECs (**Fig 5A**). However, significant differences were not observed in the total number of cellular microtubules formed between saline-treated and *ΔhlyD* mutant-infected BECs (**Fig 5B**), whereas the number was greatly reduced in WT UPEC-infected BECs (**Fig 5A**, **yellow arrowheads**). Quantification of the microtubules in UPEC-infected BECs demonstrated that WT UPEC-infected BECs retained significantly lower amounts of microtubules than *ΔhlyD* mutant-infected BECs (**Fig 5B**). In view of the reduced number of microtubules found in BECs infected with HlyA-expressing UPEC, we wondered if this was the

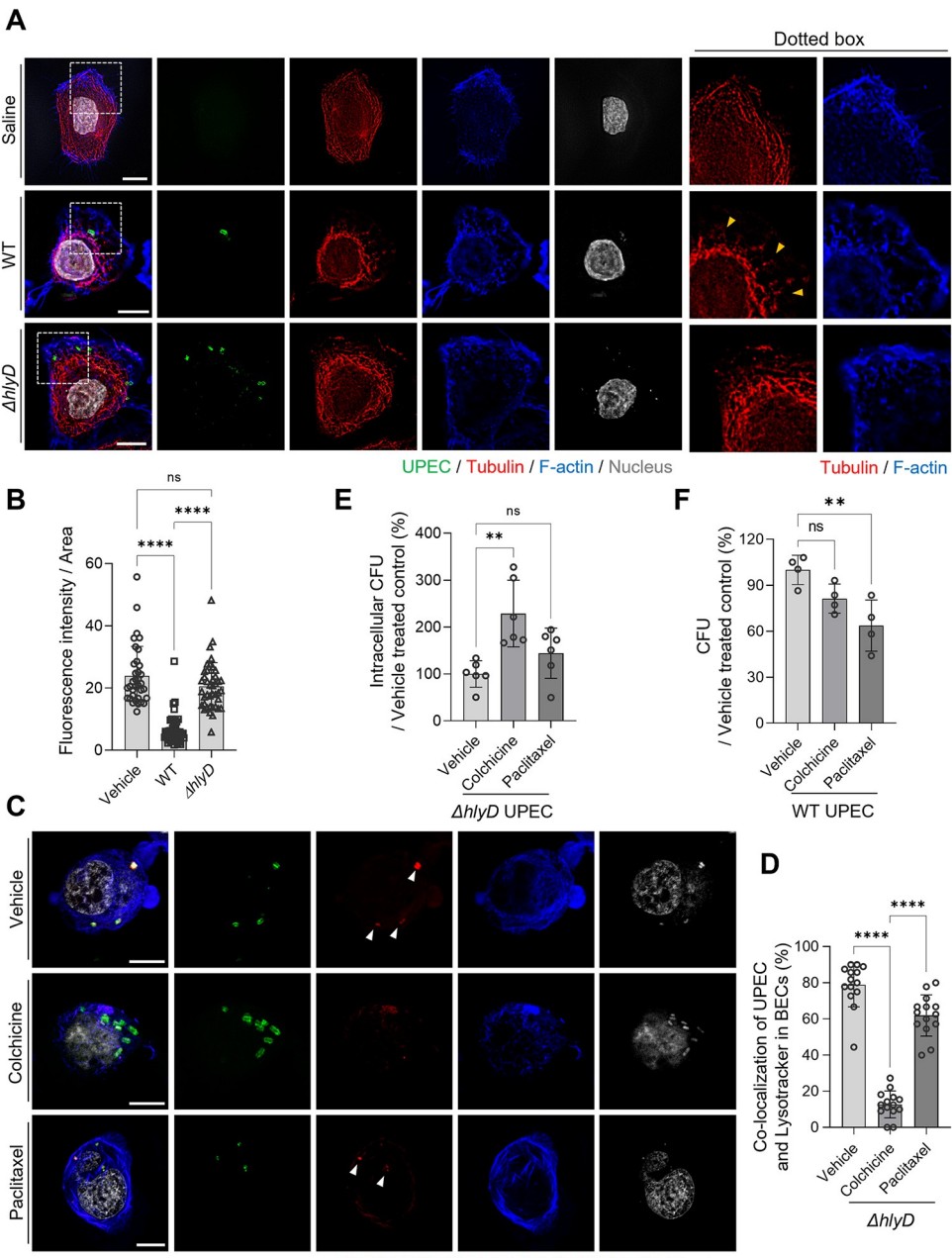

**Fig 5. HlyA-mediated microtubule disruption causes UPEC persistence. A.** HlyA disrupts the microtubule cytoskeletal structure. Human 5637 BECs were infected with WT or *ΔhlyD* UPEC CFT073 strain for 8 h and then stained with anti-tubulin (red), anti-UPEC (green), and phalloidin (blue). Dotted squares have been magnified and depicted in the right-side panels; Yellow arrowheads indicate dismantled fragments of microtubules. **B.** The optical fields were randomly selected in the microscopic images and were quantified. The mean fluorescence intensity of microtubule staining is shown in Fig 5A. **C.** The microtubule regulates the acidification of UPEC-containing vesicles. The *ΔhlyD* mutant strain-infected human 5637 BECs were treated with colchicine (10 μM) or paclitaxel (10 μM). After 24 h, the infected BECs were treated with the LysoTracke Red dye and stained with anti-tubulin (blue) and anti-UPEC (green) antibodies. **D.** The optical fields were randomly selected in the microscopic images and were quantified. The quantification of UPEC co-localization with Lysotrackers in BECs in Fig 5C. **E.** The destabilization of microtubules enhanced the bacterial burden on BECs. The human 5637 BECs were infected with the UPEC *ΔhlyD* strain. After removing the extracellular bacteria by gentamicin treatment, the cells were treated with colchicine (10 μM) or paclitaxel (10 μM) for 16 h and their bacterial burdens were compared. **F.** Paclitaxel treatment of UPEC-infected BECs reduced the bacterial load. The WT CFT073 strain infected-human 5637 BECs were treated with colchicine (10 μM) or paclitaxel (10 μM) for 16 h, and their bacterial burdens were compared. Data information: Quantitative data from two to three independent experiments were analyzed. Data are shown as mean ± SD. Data were analyzed by ordinary one-way ANOVA (B, D, E, F). **P<0.01; ****P<0.0001; n.s, not significant. Scale bar: 10 μm.

underlying reason for the lack of association of V-ATPase with bacteria-encapsulating vesicles and their subsequent lack of acidification. To address this, we examined the relationship between cellular microtubule formation, acidification of UPEC-containing vesicles, and intracellular bacterial survival in BECs. We used infected BECs treated with known agents that either disrupt microtubule formation or stabilize it, such as colchicine (10 μM) and paclitaxel (10 μM), respectively. We also employed the *ΔhlyD* mutant as the infecting bacterial strain because it does not disrupt microtubule formation in infected cells. To monitor the acidification of bacteria-harboring vesicles, we applied an acidotropic dye. Initially, we performed microscopy to examine the effects of colchicine and paclitaxel on *ΔhlyD* mutant-infected BECs. We discovered that vehicle-treated BECs contained a considerable amount of intact microtubules and that all intracellular bacteria detected in BECs co-localized with the acidotropic dye (**Fig 5C, white arrowheads**); a similar observation was made in BECs infected with the *ΔhlyD* mutant following paclitaxel treatment, where robust amounts of highly stabilized microtubules were evident (**Fig 5C, white arrowheads**). In contrast, the number of microtubules in colchicine-treated and infected BECs was markedly reduced and disrupted (**Fig 5C**). Furthermore, very few intracellular bacteria co-localized with the acidotropic dye in these cells (**Fig 5C**). Next, we sought to quantify the level of co-association between internal bacteria and the dye in all three groups. In comparison to untreated BECs, where the percentage of intracellular *ΔhlyD* mutant co-associated with acidotropic dye was as high as 80%, the number of bacteria co-associated with the acidotropic dye was minimal (less than 10%) in infected BECs treated with colchicine, indicating that microtubule formation is critical for acidification of bacteria-harboring lysosomes (**Fig 5D**). Given that we observed a limited but significant reduction in the percentage of intracellular bacteria co-associated with an acidotropic dye in BECs subjected to a microtubule stabilizer, it is conceivable that artificially stabilizing the microtubules in infected BECs may also interfere with bacterial acidification within the lysosomes (**Fig 5D**). However, paclitaxel showed no additional effects.

Next, we investigated whether the destabilization of microtubules and the resulting lack of acidification affected the intracellular survival of *ΔhlyD* mutants in BECs. Compared to vehicle- or paclitaxel-treated infected BECs, colchicine-treated infected BECs had significantly reduced intracellular bacterial killing (**Figs 5E and S19**). Given that the disruption of microtubules resulted in reduced intracellular bacterial killing in lysosomes, we hypothesized that stabilizing microtubules in BECs infected with WT HlyA-expressing UPEC using drugs such as paclitaxel would increase intracellular bacterial killing. Indeed, when we tested this notion, we found that treating CFT073-infected BECs with paclitaxel resulted in a marked reduction in the intracellular bacterial numbers, whereas treating infected BECs with colchicine had no significant effect on the intracellular bacterial count (**Figs 5F and S20**). These findings suggested that disruption of microtubule formation within infected BECs, as is the case with HlyA-expressing UPEC, promotes intracellular bacterial survival, by impeding acidification of bacteria-harboring lysosomes. However, the application of a stabilizer of microtubules, such as paclitaxel, has the potential to reverse the effects of HlyA.

## Essential role of HlyA in promoting *in vivo* UPEC persistence in the mouse bladder

Having observed the role of HlyA in the escape of UPEC from RAB27b compartments into the cytosol and subsequently promoting intracellular persistence after the bacteria were recaptured and trafficked into lysosomes in *in vitro* assays, we sought to investigate the *in vivo* relevance of these virulence traits. Prior to conducting *in vivo* studies, we confirmed the critical role of HlyA in promoting UPEC persistence inside BECs by comparing the persistence capacities of

the WT CFT073 strain and the *ΔhlyD* mutant in BECs using a competitive infection assay. To determine whether the expression of HlyA provided a competitive edge to UPEC, we infected the same BECs with equal numbers of the WT CFT073 and *ΔhlyD* mutant strains, and then compared their relative capacities to survive within the same BECs. We differentiated the bacterial strains during culture by the distinct antibiotic markers that they carried. WT CFT073 strain exhibited intracellular persistence in BECs similar to that seen previously for strain CI5 (**Fig 1D**), whereas the *ΔhlyD* mutant exhibited a limited capacity to persist within BECs (**Fig 6A**, **left panel**). Although up to $1 \times 10^3$ CFU of WT CFT073 survived within BECs by 18 h p.i., only approximately 50 CFU *ΔhlyD* mutant were detectable at the same time-point (**Fig 6A, left panel**). When the same data were presented as the percentage of total bacteria, we found that at 18 h p.i., the WT CFT073 strain made up over 95% of the total intracellular bacteria, while the *ΔhlyD* mutant made up less than 5% (**Fig 6A, right panel**). Subsequently, we performed an *in vivo* assay in which we infected mouse bladders with an equal number ($1 \times 10^8$ CFU) of the WT CFT073 and *ΔhlyD* mutant strains. We found that up to $1 \times 10^3$ CFU of CFT073 were still detectable in the bladders on day 9 p.i., whereas the *ΔhlyD* mutant had been cleared (**Fig 6B**, **left panel**). When we presented the same data as the percentage of total bacteria surviving in the bladder, it was observed that at 9 d p.i., close to 100% of the UPEC surviving in the bladder were WT CFT073, and very few (or none) *ΔhlyD* mutant survived (**Fig 6B, right panel**). Complementation with a plasmid encoding *phlyD* increased survival in BECs and mouse bladders in comparison to the *ΔhlyD* mutant strain (**S21 Fig**). These data suggested that HlyA expression by UPEC is a major contributor to their survival in the bladder. Given that HlyA contributes to intracellular bacterial survival by preventing acidification of bacteria-harboring lysosomes via disruption of cellular microtubules, we examined whether intravesicular administration of the microtubule stabilizer, paclitaxel, would impede intracellular survival. Paclitaxel treatment significantly reduced the burden of HlyA-expressing CFT073 UPEC in infected bladders (**Fig 6C**). These results suggest that the virulence attributes of HlyA can be abrogated *in vivo* by promoting the acidification of LAMP1$^+$ vesicles that house intracellular UPEC.

## Discussion

Previous UTI models in mice have revealed the capacity of UPEC to persist in bladders following acute infection for periods exceeding three weeks and long after bladder inflammation has subsided [40,41]. The persistence capacity of UPEC in the bladder has gained attention, as it could be the insidious basis for UTI recurrence in patients. Moreover, UPEC persistence in the bladder is a unique phenomenon, as it is not observed in the kidneys, even after repeated bouts of UTI [41]. Hence, our interest in investigating the basis for UPEC persistence in the mouse bladder following UTI and within cultures of human BECs *in vitro*.

Our results have revealed that UPEC persistence within human BECs is closely associated with their encapsulation within LAMP1$^+$ lysosomal compartments that failed to acidify even after prolonged incubation. This capacity of UPEC to prevent acidification of LAMP1$^+$ lysosomes was linked to the specific expression of HlyA, as LAMP1$^+$ compartments harboring the *ΔhlyD* mutant but not WT UPEC were found to become acidic and were associated with the loss of bacterial viability. Importantly, the abrogation of acidification in lysosomal compartments harboring WT UPEC correlated with the limited recruitment of V-ATPase, a proton pump that serves to acidify lysosomes. Acidification of phagosomes is achieved by the trafficking of V-ATPase proton pump-containing vesicles toward them. The lysosomal proteinases found in lysosomes and V-ATPase are typically found in distinct compartments, but when bacteria bearing vesicles become acidified, they acquire V-ATPase through the fusion of

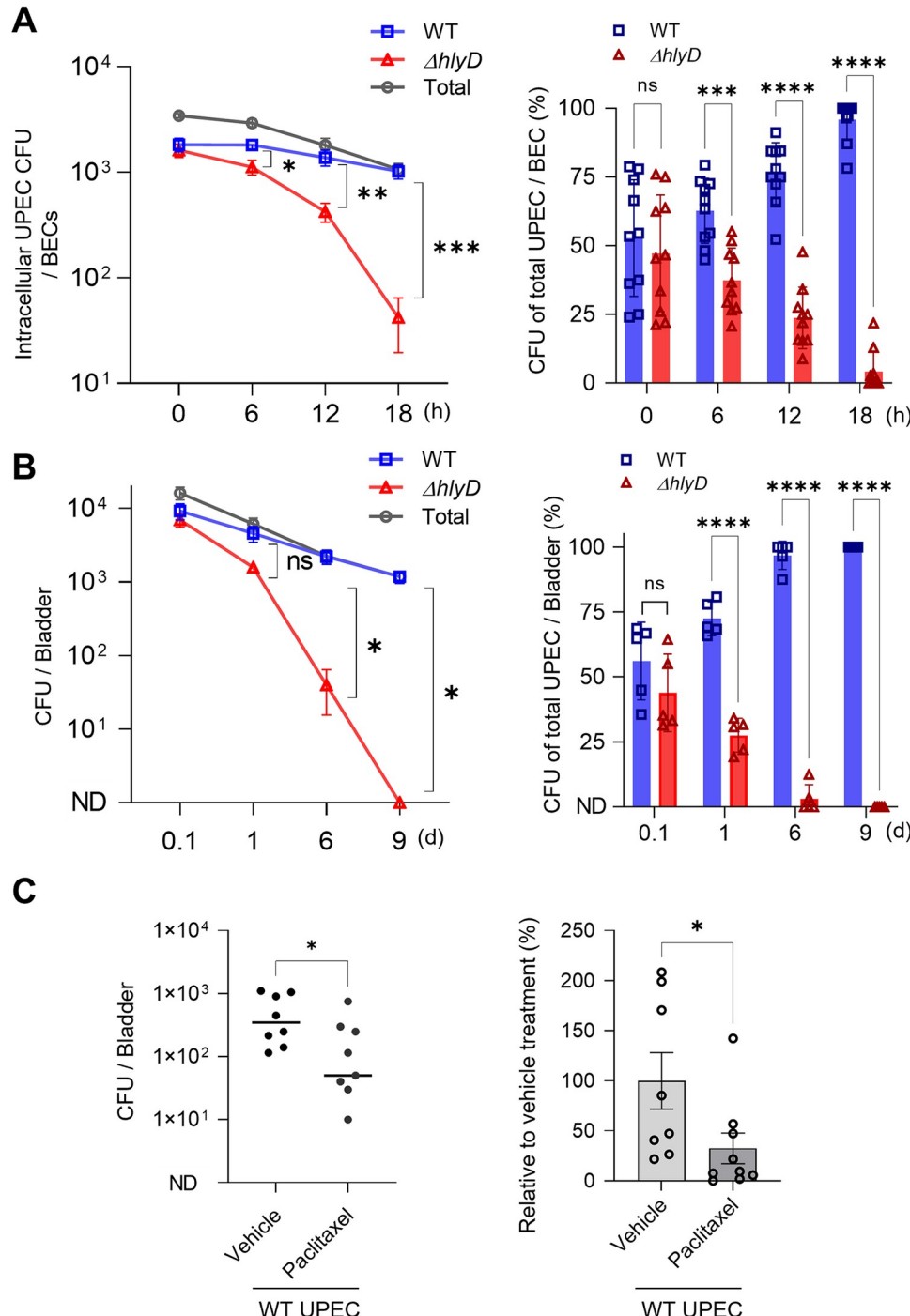

**Fig 6. HlyA is responsible for UPEC persistence in the bladder. A.** HlyA enhanced the survival of UPEC in BECs. Intracellular bacterial burden was compared after co-infecting human 5637 BECs with a 1:1 mixture of the WT CFT073 strain and *ΔhlyD* strain. (Left) Intracellular UPEC numbers. (Right) Percentage of total intracellular UPEC. **B.** HlyA enhanced the UPEC burden in the mouse bladders. The bacterial burden was compared over a time-course, after co-infecting mouse bladders (C57BL/6J) with a 1:1 mixture of the WT CFT073 strain and *ΔhlyD* mutant. (Left) Bacterial burden in the infected bladder. (Right) Percentage of total UPEC over 9 days p.i. **C.** The decreased bacterial burden after paclitaxel treatment in UPEC-infected bladders. WT mice were infected with the WT UPEC CFT073 strain. After 12 h, the bladders were treated with paclitaxel (100 μg), following which two additional treatments were given, one day apart. The bacterial burden was measured one day after the last treatment. Data information: Quantitative data from two to three independent experiments were analyzed. Fig 6A and 6B (right panels) are shown as mean ± SD and rest of the results are shown as mean ± SEM. Data were analyzed by two-way ANOVA (A, B), or unpaired two-tailed Student's *t*-test (C). *P<0.05; **P<0.01; ***P<0.001; ****P<0.0001; n.s, not significant.

V-ATPase vesicles [42,43]. Since these V-ATPase proton pumps are typically trafficked between vesicles along microtubules [44–46], disruption of microtubules in infected BECs by HlyA could explain why the compartments harboring WT UPECs fail to acidify, resulting in bacterial survival within the compartments harboring the *ΔhlyD* mutant. Moreover, while the BECs harboring the *ΔhlyD* mutant displayed minimal disruption of their tubular architecture, there was marked disruption of this architecture in BECs harboring WT UPEC. The treatment of the BECs harboring the mutant with paclitaxel, a microtubule stabilizer, promoted acidification of the bacteria-harboring vesicles, leading to loss of bacterial viability; this further validates that the loss of microtubule organization is a critical determinant of bacterial persistence within BECs. Conversely, when the *ΔhlyD* mutant-harboring BECs were treated with colchicine, a disruptor of microtubules, reduced acidification of the vesicles and enhancement of intracellular bacterial viability was observed. It is noteworthy that paclitaxel is a well-known anti-cancer drug and therefore could potentially be employed as a therapy against UTIs. We found that transurethral administration into UPEC infected mouse bladders significantly reduced the UPEC load in the bladder. Although paclitaxel has been reported to evoke side effects such as rash, hives, itching, swelling, etc. [47] in cancer patients, it is unlikely that this drug will cause similar side effects in UTI patients if it is administered transurethrally.

Production of HlyA by UPEC has been previously reported to contribute to significant cytotoxicity observed in the superficial epithelium lining the bladder during the early and acute phases of infection [16,23–25]. In addition to the extracellular secretion of HlyA by UPEC, the contribution of intracellularly secreted HlyA and their regulation are crucial to our understanding of HlyA-mediated pathogenesis [48,49]. In this study, we observed that HlyA is also essential for establishing a persistent or chronic UPEC infection. The critical role of HlyA in promoting bacterial persistence was validated by the results obtained from our mouse UTI model, which showed that after co-infection of the bladder with comparable numbers of WT and *ΔhlyD* mutant bacteria, by day 9 p.i., all surviving bacteria detected in the bladder were WT UPEC. Presumably, the *ΔhlyD* mutant cannot persist without the capacity to neutralize LAMP1+ vesicles within BECs. Microtubule stabilization within BECs is essential for eliminating the UPEC from the mouse bladder, which is proved by the fact that intravesical instillation of paclitaxel could significantly reduce bacterial persistence in the bladder. Although we have not shown how HlyA disrupts cellular microtubules or enters the cytoplasm of the host to dismantle the cytoskeleton, HlyA is known to indirectly disrupt the cytoskeletal structure by inducing the degradation of paxillin, which reorganizes the actin cytoskeleton and is essential for cell adhesion, spreading, and migration [25,50,51]. It has also been suggested that paxillin degradation is an early initiating event in the HlyA mediated cytotoxicity of the superficial epithelium mentioned earlier [25]. HlyA expressing UPEC possess the remarkable ability to cause significant cytotoxicity and inflammation, including robust neutrophil recruitment early in the course of infection and subsequent promotion of intracellular persistence within BECs when bladder inflammation has subsided. This could suggest an innate capacity of UPEC to tightly regulate HlyA expression. Indeed, the HlyA gene in UPEC is under the regulatory control of the CpxRA stress response system [24]. Presumably, CpxRA is responsible for up-regulating HlyA expression to achieve successful infection of the bladder, in spite of the robust innate immune response of the host, and later, when the inflammation has subsided, adjusting HlyA expression to promote intracellular bacterial persistence and possibly reemergence of the infection. Since our in vitro studies suggest that much of the trafficking of UPEC between RAB27b+ and LAMP1+ vesicles occurs in the first 8 h of UPEC infection, we suspect these events occur before the proposed QIR phase [9] and the formation of intracellular bacterial communities [15] which are other intracellular phases of UPEC existence in BECs.

Our studies have confirmed earlier studies suggesting that UPEC entry into BECs involves coopting the endocytic properties of RAB27b[+] fusiform vesicles [8]. These fusiform vesicles serve an important physiologic function by fusing to the apical surface of the superficial epithelium of the bladder with the help of VAMP8/endobrevin to facilitate bladder expansion [52]. Following entry into BECs, RAB27b encased UPEC follow a circuitous path dictated in part by the virulent traits of the bacteria and the corresponding counter measures instituted by the host cell. Our results reveal that the early cytosolic escape of UPEC from RAB27b[+] vesicles is critically mediated by HlyA. We found that at 5 h p.i., while most intracellular *ΔhlyD* UPEC were still associated with RAB27b[+] vesicles, very few of the WT UPEC remained associated with the fusiform vesicles. This HlyA-mediated escape into the cytosol appeared to be beneficial to UPEC, as the intracellular counts of WT UPEC in BECs were significantly higher than the corresponding numbers of *ΔhlyD* UPEC. It is not known why HlyA-producing *E. coli* escape RAB27b[+] compartments and enter the cytosol, whereas the same does not occur when the same bacteria are harbored in LAMP1[+] lysosomes. We suspect that the heterogeneous $Ca^{2+}$ concentrations in the vesicles are directly associated with HlyA activity. Biochemical analysis of HlyA showed that the RTX domain of HlyA possesses six $Ca^{2+}$-binding sites essential for folding, which are responsible for its biological activity [53]. The secreted HlyA in the early endosome, which retains a similar level of $Ca^{2+}$ concentration as the cytoplasm, may have membrane lytic activity [54]. Therefore, the $Ca^{2+}$-bound active form of HlyA enables UPEC to escape from the RAB27b[+] vesicles into the cytoplasm. However, $Ca^{2+}$ levels rapidly deplete during endosomal vesicle trafficking, which may hinder the formation of a fully active complex, owing to a deficiency of $Ca^{2+}$ in HlyA [54]. Therefore, it is conceivable that HlyA activity within RAB27b[+] and lysosomal vesicles may be regulated by the heterogeneous integrity of their $Ca^{2+}$ concentrations. The release of HlyA producing UPEC from RAB27b vesicles into the cytosol without impacting cell viability raises the question of why vesicular membranes appear more susceptible to HlyA than the plasma membrane. Conceivably, the plasma membrane is more resistant to HlyA than the vesicular membrane because of differences in their cholesterol composition or in membrane-embedded proteins [55,56]. It must be emphasized that many of the findings described here have involved the use of a human bladder epithelial cell line, which lacks some of the attributes of primary epithelial cells such as the formation of plaques on the apical surface of urothelial cells [12,57,58]. Thus, there is a need to validate our current findings in physiologically more relevant primary bladder epithelial cells or human urothelial organoid [59].

Finally, our study reveals a distinct virulence activity mediated by HlyA produced by UPEC strains that appears to promote bacterial persistence in the bladder. This finding is in agreement with the finding that the expression of HlyA among UPEC isolates from recurrent UTI patients is markedly higher than isolates from first- or second-time UTI patients [26]. The strategy employed by UPEC to prevent the acidification of lysosomes is not unlike the tactics utilized by well-known intracellular pathogens such as *Salmonella* spp. and *Mycobacteria* spp. which prevent the fusion of bacteria-harboring vesicles with lysosomes containing degradative enzymes [60–62]. In view of the capacity of UPEC to promote bacterial persistence in the bladder through the production of HlyA, this virulence factor could be a vaccine candidate, as vaccines are being contemplated to combat recurrent UTIs. Evoking HlyA targeting antibodies and cytotoxic T cells could potentially be highly effective in abrogating the cytotoxic actions of UPEC during the early phases of infection and in preventing the long-term persistence of UPEC in BECs. Currently, patients experiencing recurrent UTIs are often subjected to prolonged antibiotic treatment, which has side effects such as toxicity to major organs and the emergence of antibiotic resistance. Our finding that stabilization of cellular microtubules within the bladder is sufficient to promote clearance of persistent bacteria in the bladder

highlights a promising new strategy for the treatment of UTIs. Conceivably, combining antibiotic treatment with transurethral administration of paclitaxel may have a synergistic effect in achieving clearance of HlyA$^+$ UPEC from the infected bladder.

## Materials and methods

### Ethics statement

The animal-associated experimental procedures were performed with the approval of the Korea University Institutional Animal Care & Use Committee. Bladder biopsies were taken from recurrent UTI patients who had cystoscopy-confirmed IC/BPS. The human bladder sample study was approved by the Institutional Review Board at Ewha Womans University Seoul Hospital. All human subjects provided informed written consent prior to participating in this study. Urine was cultured from these patients to establish the presence of uropathogens.

### Bacterial strains

UPEC CI5 [63–65], J96, and CFT073 wild-type [16] were used for infection as HlyA$^+$ UPEC strain. CFT073 *ΔhlyD* mutant strain [16] was used as HlyA$^-$ UPEC strain. CFT073 wild-type and CFT073 *ΔhlyD* mutant strains were a generous gift from Dr. Mobley at the University of Michigan in Ann Arbor. *E. coli* K12 strain MG1655 sub-strain was employed as nonpathogenic bacteria control. All the strains were statically grown in Luria–Bertani broth for 16–18 h at 37˚C. Bacterial colonies were counted by plating on MacConkey agar plates with an overnight incubation at 37˚C.

### Bladder cell line and mice

Human 5637 bladder epithelial cells (ATCC, HTB-9) were grown in RPMI 1640 (Gibco) with 10 mM HEPES, 1 mM sodium pyruvate, 10% FBS (Sigma) and 0.225% glucose in a 5% CO$_2$, 37˚C incubator. Eight- to ten-week-old female C57BL/6J mice were purchased from Jackson Laboratories.

### *In vivo* mouse infection, CFU count assay, and paclitaxel administration

Anesthetized mice were inoculated transurethrally with $1 \times 10^8$ UPEC, using the indicated UPEC strains. The infected mice were sacrificed at the desired times post-infection. While examining the therapeutic effect of paclitaxel (Sigma), 5 mg/kg of paclitaxel dissolved in 50% polyethylene glycol (PEG) 400 solution (made in 0.9% saline solution) was administered intravesically three times with a one-day interval between administrations. The bladders were dissected and suspended in PBS, followed by homogenization using a bead-beating homogenizer. The UPEC CFU counts in the homogenized bladders were obtained by plating diluted homogenate on MacConkey agar plates for overnight incubation at 37˚C. To further demonstrate the bacterial burden in the bladder, the in vivo results were expressed in CFU/mL unit as shown in **S22 Fig**. The detection limit of CFU in mouse bladders was 100 CFU per bladder.

### Human tissue biopsy and immunostaining

Human bladder biopsies were processed following formaldehyde fixation. Tissues-embedded in paraffin were sliced at a thickness of 5 μm and deparaffinized. To eliminate antigen masking caused by formalin, heat-mediated antigen retrieval in sodium citrate buffer was conducted. After blocking the tissue slices with a solution comprising 10% donkey serum, 0.35% Triton X-100, and 1% bovine serum albumin in PBS, the samples were treated with anti-LAMP1 (Abcam, H4A3), anti-E-cadherin (BD Biosciences, 610181), or anti-*E. coli* (Bio-Rad,

OBT0986) antibodies. Secondary antibodies linked to a fluorochrome (Jackson ImmunoResearch) was utilized to visualize the target.

## Antibodies and chemical reagents

Antibodies: anti-LC3A/B (Cell Signaling, 4108), anti-ATG5 (Cell Signaling, 2630), anti-BECLIN1 (Novus Biologicals, NB500-249), anti-LAMP1, anti-RAB27b (IBL–America, 18973), anti-Fim H (CUSABIO, CSB-PA362349ZA01ENV), anti-Gapdh (GeneTex, GTX627408), anti-β-actin antibodies (Abcam, 8224), Anti-Endobrevin (Santa Cruz Biotechnology, Sc-166820), anti-Ly6G (BD Biosciences, clone 1A8), anti-CD11b (BD Biosciences, clone M1/70), or anti-CD45 (BD Biosciences, clone 30-F11) antibodies. Densitometry graphs showing relative changes in protein expression normalized to control were added in **S23 Fig**. Reagents: Gentamicin (Gibco, 15750–078), Paclitaxel (Sigma-Aldrich, T7191), Colchicine (Sigma-Aldrich, C9754), Bafilomycin A1 (LC Laboratories, B-1080), or nucleus staining (DAPI containing Prolong Diamond Antifade mounting solution, Thermo Fisher Scientific).

## Immunofluorescence microscopy

Human 5637 BECs were seeded on glass coverslips at approximately $1 \times 10^5$ cells/well in tissue culture-treated 24 well plates for a period of 16 h. Seeded cells were infected with different UPEC strains at MOIs ranging from 20–400 for a period of 1 h. After infecting the cells, gentamicin (200 μg/ml) and methyl D-mannose (0.5%) were added to the media and incubated for 30 min to remove the extracellular UPECs. D-mannose to media prevents UPEC from reattaching and entering the BECs. Prolonged incubation was performed in the media containing gentamicin (50 μg/ml) and methyl D-mannose (0.5%). The cells were fixed with 4% paraformaldehyde at relevant time points in the experiment for 20 min at RT. The fixed cells were permeabilized and blocked with 0.1% saponin in 1% BSA for 1 h. The blocked cells were treated with a primary antibody and followed by appropriate fluorophore-conjugated secondary antibodies. The examination of slides was performed using a Zeiss LSM800 confocal microscope or Lecia Thunder imager. The processing of images or quantification was determined using the image processing software Image J (National Institutes of Health).

## Intracellular CFU count assay

$2 \times 10^5$ human 5637 BECs were seeded in a well of 24-well plates for 16–20 h. UPEC was added to each well at the intended MOI, followed by centrifugation for 4 min at 1200 rpm. After 1 h of infection, the medium was replaced with fresh media containing gentamicin (200 μg/ml) for 30 min. The gentamicin concentration in the media was reduced to 50 μg/ml for the remainder of the experiment. At relevant time points, the cells were washed thrice with PBS. To lyse the cells, 0.1% Triton X-100 in PBS was used, and treated cells were scraped, diluted, and plated onto MacConkey agar plates. Counted colonies represented the quantity of invaded UPEC.

## Extracellular CFU count assay

Approximately $2 \times 10^5$ human 5637 BECs were seeded per well in a 24 well plate for 16–20 h. UPEC was added to each well at the appropriate MOI, followed by a 4 min spin at 1200 rpm. After an hour of infection, the medium was replaced with 200 μg/ml gentamicin in RPMI medium for 30 min at 37˚C. Post gentamicin treatment, a bacteriostatic medium consisting of 0.5% mannose and 62.5 μg/ml sulphamethoxazole (SMZ) in RPMI medium was used and replaced every hour during the experiment. The replaced media at the end of each hour was pooled and spun at 3000 rpm for 10 min to pellet the bacteria and suspended them in PBS.

The suspension was plated on MacConkey agar for extracellular colony counts at each time point in the experiment.

## Magnetic labeling of live bacteria

Live bacteria were labeled using BioMag carboxyl magnetite particles (Bangs Laboratories, BM570). The stock suspension of particles was initially spun down at 1000 g for 1 min and 900 μL of the supernatant was used for the experiment. The BioMag particles were then washed twice with pH 5.2, 0.1 M MES buffer (2-(N-morpholino) ethanesulphonic acid) using a magnetic rack (BD IMagnet, BD Biosciences, 552311), followed by the addition of 4 mg EDAC (1-ethyl-3- (3-dimethylaminopropyl) carbodiimide). The BioMag suspension was rotated (12 rpm) in an RT incubator for 15 min to chemically activate the particles. The suspension was washed twice with PBS on the magnetic rack. The particles were then empirically mixed with live bacteria ($0.5 \times 10^9$ CFU) in a total volume of 1 ml PBS. The particle-bacteria suspension was further incubated on a rotator (12 rpm) for 30 min at 37˚C. Blocking with 1% BSA in PBS for 30 min in a rotator at RT followed. Centrifugation at 200 g for 1min allowed for the removal of the larger aggregates.

## Intracellular vesicle isolation using magnetically labeled bacteria

Human 5637 BECs were seeded in T-150 plates and incubated at 37˚C till 100% confluency was achieved. UPECs were magnetically labeled with the protocol previously mentioned. Human 5637 BECs were then infected with the magnetically labeled bacteria at the appropriate MOI for 2 h at 37˚C, followed by a 200 μg/ml gentamicin treatment for 1 h. A lower concentration of 50 μg/ml of gentamicin was applied for the remainder of the experiment. At relevant time points, the cells were trypsinized with a 0.5% trypsin solution. The pelleted cells were suspended in ice-cold homogenization buffer containing 20 mM HEPES, 0.5 mM EGTA, 0.25 M sucrose, 0.1% gelatin, and mammalian & bacterial protease inhibitors. Passing the cells through 25 G and 30 G needles 7–10 times each, on ice, lysed the BEC membrane. The lysate tubes were loaded onto a magnetic separation rack for 2–4 h at 4˚C till a brown smear was noticed on the side of the tubes facing the magnetic separation rack. The Brown smear was suspended in fresh chilled homogenization buffer and loaded onto the magnetic rack for 30 min. This step was repeated thrice, followed by the suspension of the isolated vesicles in PBS for further analysis.

## Transient knockdown or overexpression of genes in human 5637 BECs

100 pmols of siRNA targeting the desired host molecules or non-specific siRNA duplexes (Ambion) with Lipofectamine 2000 (Invitrogen) in Opti-MEM media (Invitrogen) were used to transfect $0.5 \times 10^5$ BECs. After 24–36 h, transfected cells were used for assays of interest. The knockdown was confirmed via Western blot analysis using antibodies against the target molecule. In this study the following siRNA sequences were used: si*ATG*5 Sense: 5'CAAUCC CAUCCAGAGUUGCUUGUGA 3', antisense: 5'UCACAAGCAACUCUGGAU G GGAUUG 3'); siATP6V1A (V-ATPase siRNA) Sense: 5' GAGCUUGAAUUUGAAGGUGUAdT dT -3', antisense: (5' UACACCUUCAAAUUCAAGCUCdTdT 3'). pEGFP-LC3 plasmid (Addgene, Plasmid #24920) was used to overexpress LC3B by using ScreenFect transfection reagent (Fujifilm Wako Chemicals).

## Gentamicin protection assay

UPEC was magnetically labeled using the methods described above. UPEC- housing vesicles from infected human BECs were isolated at desired times post-infection using the previously

mentioned protocol. The isolated vesicles were suspended in 200 μg/ml gentamicin containing PBS for 30 min at 37˚C. The solution was spun at 3000 g to pellet the UPEC, followed by suspension of the pellet in PBS and plating on MacConkey media plates. The positive control consisted of treating the isolated vesicles with 200 μg/ml gentamicin and 0.1% Triton X100 in PBS solution for 30 min at 37˚C.

## Hemolysis assay

Red blood cells (RBCs) were isolated from blood donations. RBCs were suspended in an ACD (acid-citrate-dextrose) buffer containing 39 mM citric acid, 75 mM sodium citrate, and 135 mM dextrose at pH 7.4. In a 1:1 ratio, the RBC suspension was combined with the supernatant of UPEC strains grown overnight in LB broth. The combined suspension was incubated at 37˚C for 120 min while shaking (250 rpm). This was followed by a 10min, 1000 g spin at 4˚C. The optical density (OD) at 540 nm was determined for the supernatants using a spectrophotometer. Treatment with 0.1% Triton X100 was used as the complete hemolysis for the calculation of the percentage.

## Cytotoxicity assay

To analyze the cytotoxicity of UPEC strains on human 5737 BECs, the CytoTox 96 non-radioactive cytotoxicity assay kit (Promega) was used. Human 5637 BECs were seeded in 96-well cell culture plates. The following day, UPEC strains were spun at 300 g to treat them. After one hour of incubation, gentamicin-containing medium (1 μg/ml) was used for additional culture. The release of lactate dehydrogenase from the culture supernatant was measured after 16 h of incubation. Lysed BECs were used to determine the percentage of total lactate dehydrogenase released into the supernatant [66].

## Complementation of *hlyD* gene

Full-length *hlyD* was amplified from CFT073 strain and cloned into pWSK29 plasmid. The *hlyD* cloning primer sequences: forward 5' GCCGGATCCATGAAAACATGGTTAATGGGG 3', rear 5' CGGCCTCGAGTTAACGCTCATGTAAACTTTC'. Then, *ΔhlyD* CFT073 strain was transformed by electroporation with 100 μg/ml carbenicillin for selection to generate *ΔhlyD*(*phlyD*) UPEC strain.

## Quantification and statistical analysis

The data was analyzed, and graphics were created using Prism (GraphPad Software). P values less than 0.05 were considered significant statistically. Each graph (*in vitro* experiments) has dots that represent samples from two to three independent experiments. To determine statistical significance, we utilized the unpaired Student's t-test, the one-way ANOVA with Tukey's multiple comparisons test, or Kruskal-Wallis test. The dots on each graph (mouse experiments) indicate the number of mice used to obtain bladders. The two-way ANOVA with Tukey's multiple comparisons test was used to compare the bar graphs as described in the figure legends.

## Supporting information

**S1 Fig. Bacterial burden in the infected mouse bladders at 8 weeks of post infection.** C57BL/6 female mice were infected by intravesical instillation of the UPEC CI5 strain and bacterial CFUs in the infected bladders were measured at 8 weeks post-infection. (TIF)

**S2 Fig. A gating strategy to define the neutrophil population in Fig 1C.** Ly6G⁺ CD11b⁺ CD45⁺ cells in the mouse bladders were counted by flow cytometry.
(TIF)

**S3 Fig. Presence of persistent UPEC in human bladder tissue.** Bladder biopsies were obtained from recurrent UTI patients or patients with no history of UTI in past two year. These tissues were immunostained for UPEC (E. coli, green), urothelium (E-cadherin, blue), lysosomal vesicle (LAMP1, red), and nucleus (DAPI, gray). Yellow arrowheads indicate UPEC. "L": lumen of the bladder. Scale bars: 10 μm.
(TIF)

**S4 Fig. UPEC localization in LC3 autophagosome without the presence of RAB27b.** (A) Human 5637 BECs were infected with UPEC CI5 strain for 8 h and were stained with anti-LC3A/B (red) and anti-UPEC (green) antibodies. (B, C) Human 5637 BECs were transfected with pEGFP-LC3 (green) to specifically trace LC3 autophagy component. Next day, the cells were infected with UPEC CI5 strain. After 4 or 8 h p.i., cells were stained with anti-*E. coli* (red) and anti-RAB27b (blue) antibodies. Representative images are from randomly selected regions. Scale bars: 10 μm.
(TIF)

**S5 Fig. No significant difference in the intracellular bacterial numbers of labeled and non-labelled bacteria.** Human 5637 BECs were treated with magnetic bead-labeled UPEC CI5 strain or UPEC CI5 strain (no labeled beads). After 2 h of incubation with gentamicin to remove extracellular UPEC, intracellular CFU were examined. ns: not significant.
(TIF)

**S6 Fig. Isolated magnetic bead-labeled UPEC still express FimH and are associated with Rab27⁺.** Human 5637 BECs were incubated with magnetic beads alone or with magnetic bead-labeled UPEC. After two hours of incubation, isolated beads or cell lysates from each condition were analyzed for the expression of RAB27b, Endobrevin/VAMP8, and FimH. Gapdh was used for loading control of UPEC-containing vesicle.
(TIF)

**S7 Fig. Intracellular vesicle isolation using magnetically labeled bacteria.** Human 5637 BECs were seeded and infected with magnetically labeled UPECs. After lysing cells by passing the cells through needles, UPEC-containing vesicles were isolated with magnet. Isolated vesicles containing UPECs were treated with 0.1% Triton X-100 or saline, then incubated with gentamicin before plating on McConkey agar plates.
(TIF)

**S8 Fig. CFU count in absolute numbers for Fig 2E.** Data were analyzed by two-way ANOVA. *P<0.05
(TIF)

**S9 Fig. Decreased expression of ATG5 upon the siRNA transfection.** The 5637 BECs transfected with NS siRNA or ATG5 siRNA and immunoblotted using an anti-ATG5 antibody to examine the expression of ATG5. Anti-β-actin antibody was used for a loading control.
(TIF)

**S10 Fig. CFU count in absolute numbers for Fig 2G.** Data were analyzed by two-way ANOVA. *P<0.05
(TIF)

**S11 Fig. CFU count as absolute numbers for Fig 2H.** Data were analyzed by one-way ANOVA. *P<0.05
(TIF)

**S12 Fig. Escape of ΔhlyD UPECs from RAB27b vesicles after complementation.** *ΔhlyD* UPEC strain was transformed with *phlyD* plasmid to generate *ΔhlyD*(*phlyD*) strain. Human 5637 BECs were infected with magnetic beads labeled *ΔhlyD*(*phlyD*) strain or *ΔhlyD* with empty plasmid. After 5 h of infection, isolated UPEC-containing vesicles were analyzed for the expression of RAB27b. Gapdh was used for loading control of UPEC-containing vesicles.
(TIF)

**S13 Fig. CFU count as absolute numbers for Fig 3F.** Data were analyzed by one-way ANOVA. *P<0.05
(TIF)

**S14 Fig. CFU count as absolute numbers for Fig 3G.** Data were analyzed by one-way ANOVA. *P<0.05
(TIF)

**S15 Fig. CFU count as absolute numbers for Fig 3H.** Data were analyzed by two-way ANOVA. *P<0.05
(TIF)

**S16 Fig. Deacidification of *ΔhlyD* UPECs after complementation of *hlyD* gene.** *ΔhlyD* UPEC strain was transformed with *phlyD* plasmid to generate *ΔhlyD*(*phlyD*) strain. Human 5637 BECs were infected with *ΔhlyD*(*phlyD*) strain or *ΔhlyD* with empty plasmid. After 24 h of infection, BECs were dyed with LysoTracker (red) to trace the acidification of UPEC-containing vesicles. Anti-UPEC (green) and anti-LAMP1 (blue) antibodies were used to stain the fixed cells. Quantitative data from two independent experiments were analyzed. Data are shown as mean ±SD. Data were analyzed by unpaired two-tailed Student's t-test. ****P<0.0001. Scale bar: 10 μm.
(TIF)

**S17 Fig. Decreased expression of V-ATPase upon the siRNA transfection.** The 5637 BECs transfected with NS siRNA or V-ATPase siRNA (25, 50, or 100 pmol) and immunoblotted using an anti-V-ATPase antibody to examine the expression of V-ATPase. Anti-β-actin antibody was used for a loading control.
(TIF)

**S18 Fig. Pharmacological inhibition of V-ATPase.** Human 5637 BECs were infected with UPEC *ΔhlyD* strain, and extracellular UPEC was removed by gentamycin treatment. Then, infected BECs were incubated in media containing Bafilomycin A1, a known pharmacological inhibitor of V-ATPase (vacuolar H$^+$-ATPase), at 100 nM concentration. After 24 h post-treatment, LysoTracker Red dye was applied and then stained with anti-LAMP1 (blue) or anti-UPEC (green) antibodies for confocal microscopy imaging. The number of UPEC co-localized with LysoTracker Red dye was quantified in randomly chosen fields. Data were analyzed by unpaired two-tailed Student's t-test. ****P<0.0001. Scale bar: 10 μm.
(TIF)

**S19 Fig. CFU count as absolute numbers for Fig 5E.** Data were analyzed by an ordinary one-way ANOVA (B, D, E, F). **P<0.01; n.s, not significant.
(TIF)

**S20 Fig. CFU count as absolute numbers for Fig 5F.** Data were analyzed by an ordinary one-way ANOVA. **P<0.01; n.s, not significant.
(TIF)

**S21 Fig. Recovery of UPEC persistency after complementation on *ΔhlyD* UPEC strain.**
*ΔhlyD* UPEC strain was transformed with *phlyD* plasmid to generate *ΔhlyD*(*phlyD*) strain. (A) Human 5637 BECs were co-infected with equal numbers of the *ΔhlyD* and *ΔhlyD*(*phlyD*) strains, and intracellular UPEC numbers were examined at indicated time points. (B) After infection with these UPEC strains on human BECs, the cells were stained with anti-tubulin (red), anti-UPEC (green), and anti-LAMP1 (blue) antibodies for confocal microscopic imaging. (C) C57BL/6J was co-infected with equal numbers of the *ΔhlyD* and *ΔhlyD*(*phlyD*) strains and bacterial burden in mouse bladders were examined. Data were analyzed by two-way ANOVA (A, C). *P<0.05; **P<0.01, Scale bar: 10 μm.
(TIF)

**S22 Fig. The mouse bacterial burden in the bladder was presented as colony-forming units per milliliter (CFU/mL).** The detection limit of CFU in mouse bladders was 100 CFU per bladder.
(TIF)

**S23 Fig. Densitometry of western blot data.** Representative results from two to three independent western blots were selected and presented.
(TIF)

## Acknowledgments

We thank H. Mobley for providing the UPEC strains. V. Friedrich provided critical manuscript review.

## Author Contributions

**Conceptualization:** Manisha Naskar, Viraj P. Parekh, Soman N. Abraham.

**Formal analysis:** Manisha Naskar, Viraj P. Parekh, Mathew A. Abraham, Zehra Alibasic, Min Jung Kim, Gyeongseo Suk, Joo Hwan Noh, Kwan Young Ko, Joonha Lee, Hae Woong Choi.

**Funding acquisition:** Soman N. Abraham, Hae Woong Choi.

**Investigation:** Hae Woong Choi.

**Resources:** Hana Yoon.

**Supervision:** Hae Woong Choi.

**Writing – original draft:** Hae Woong Choi.

**Writing – review & editing:** Mathew A. Abraham, Chungho Kim, Hana Yoon, Soman N. Abraham, Hae Woong Choi.

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
