## [Decision Letter · Decision Letter 0]

25 Jul 2022

Dear Professor Choi,

Thank you very much for submitting your manuscript "α-Hemolysin Promotes Uropathogenic E. coli Persistence in Bladder Epithelial Cells Via Abrogating Bacteria-Harboring Lysosome Acidification" for consideration at PLOS Pathogens. As with all papers reviewed by the journal, your manuscript was reviewed by members of the editorial board and by several independent reviewers. The reviewers appreciated the attention to an important problem, but raised some substantial concerns about the manuscript as it currently stands. These issues must be addressed before we would be willing to consider a revised version of your study. In light of the reviews (below this email), we would like to invite the resubmission of a significantly-revised version that takes into account the reviewers' comments.

Sincerely,

Sargurunathan Subashchandrabose

Guest Editor

PLOS Pathogens

Brian Coombes

Section Editor

PLOS Pathogens

Kasturi Haldar

Editor-in-Chief

PLOS Pathogens

orcid.org/0000-0001-5065-158X

Michael Malim

Editor-in-Chief

PLOS Pathogens

orcid.org/0000-0002-7699-2064

Reviewer's Responses to Questions

**Part I - Summary**

Reviewer #1: This manuscript from the Abraham lab describes life cycle of UPEC inside bladder epithelial cell in the context of UPEC persistence. More specifically, UPEC escape Rab27b+ vesicles to become cytoplasmic using hemolysin activity; cytoplasmic UPEC are repactured by LC3A/B+ autophagosomes; and HlyA activity prevents acidification of autophagosomes into lysosomes. The authors show that hlyAKO mutant fails to leave Rab27b+ vesicles and is expelled from the cell. These results suggest the role for hemolysin activity in persistence of UPEC inside BECs.

Reviewer #2: In this work the authors probe the intracellular dynamics of UPEC within 5637 bladder epithelial cells and the role of the UPEC HylA toxin in preventing autolysosome acidification. The authors further postulate that HylA functions to reduce V-ATPase recruitment to the autolysosome by destabilizing microtubules. The authors go on to test the effects of intravesical instillation of a microtubule stabilizer to reduce bladder bacterial burden. While this work is interesting, I do have some concerns.

1. The introduction and discussion need extensive revision for clarity, grammar, and inclusiveness of the existing literature. There are several existing papers about UPEC hemolysin A that, for example, have not been referenced or adequately discussed. These include two papers https://doi.org/10.1073/pnas.150037411 and

https://doi.org/10.1128/IAI.00075-08 that report that loss of HylA does not affect bladder colonization in a mouse model. The first reference specifically shows that loss of HylA did not affect intracellular populations of UPEC or IBC formation in a mouse UTI model. Also, although doi: 10.1016/j.chom.2011.12.003 is referenced by single sentence in the discussion, since this work was the first to report microtubule disruption by UPEC HylA (via paxillin degradation) a more thorough discussion is warranted. The authors are encouraged to more thoroughly contextualize their findings within the existing literature.

2. I would encourage the authors to report absolute numbers of bacteria instead of just as percentages. These data can be included in the supplement. With only the percentage of UPEC in each compartment reported, for example it is impossible to know if the absolute numbers of bacteria in different compartments actually changes as the infection progresses (for example, the same number of bacteria could be in Rab27 vesicles at later time points but appear less as percentage if there are more total intracellular UPEC at later time points). In general, all data should be reported as absolute numbers so changes in populations can be more accurately assessed by the reader.

Reviewer #3: This manuscript builds on prior work investigating UPEC persistence within bladders. Persistent/chronic UTI has been investigated by several groups, including a possible role for HlyA hemolysin. However, this is the first to state a role for the HlyA hemolysin in lysosome acidification. The premise is interesting, but some substantial concerns need to be addressed.

A major strength is the variety of assays that were conducted, including tracking of magnetically labeled bacteria in cultured cells and treatments with reagents to reduce bacterial burden in both cultured cells and a mouse bladder model, although this is tempered by the uneven application of these assays across different experimental conditions.

There are major concerns about the lack of description for several methods, lack of some important controls, and overstated conclusions.

**Part II – Major Issues: Key Experiments Required for Acceptance**

Reviewer #1: Page 4 last paragraph: As many as 37.6% of UPEC isolated from first-time UTI patients are reported to express the hlyA gene [20] but the expression rate significantly increased to 48.2% in patients experiencing recurring UTIs, implying a role of HlyA in the recurrence of UTIs [20].— Provide sample size for the cohorts with first-time UTI and recurrent UTI. Discuss why UPEC isolated from only 31% of subjects with second UTI were hlyA+.

Please add scale bars to all microscopic images.

Circles, squares, and rectangles are used to indicate different treatments throughout the manuscript. This is confusing and must be made consistent.

Figure legends must be rewritten to include relevant information

Fig 1A and C. The legend states that the data are from two-three biological replicates. How many mice were examined in each of the biological replicates? The CFU data (Fig 1A) should be presented with median as central tendency and analyzed for statistical significance using Mann Whitney U test. Also, SEM is not the correct measure of uncertainty.

Fig 1B. >50% UPEC (red) appear to be NOT associated with LAMP1+ vesicles (blue) at 9 day pi. It will be informative if the authors quantify these data are present them.

The authors mention on page 5, “However, it is noteworthy that by day 9, no neutrophils were detected in the urine of the infected mice.” How did the authors reach this conclusion?

Fig 1C. Define the positive control for MPO assay. Also, present raw MPO data either as ΔOD/min/g of tissue OR by extrapolating it to a known MPO standard. The current presentation shows % relative MPO activity with 700% MPO activity at 6 hpi. This does not make sense. How many biological replicates do these data represent? Why is the sample size at 6 hpi so low?

Figure 2 legend states that the data are from two-three biological replicates. How many technical replicates in each biological replicate? Also, SEM is not the correct measure of uncertainty. Replace it with SD. Please do this for all relevant results presented in Fig 2 (panels B, C, G, H) and other figures.

Provide details for confocal microscopy quantification data in Fig 2B, C, how many total infected cells did you count to reach the % of UPEC associated with Rab27b+ vesicles? Is it correct to say that in the presented image (Fig 2A) only one out of three cells is infected?

Fig 2H shows that siRNA treatment results in 200% of infected cells to contain UPEC. What does this mean? Also, are these UPEC associated with a specific vesicle?

The authors should explicitly state that Fig 2 shows in vitro experiments using 5637 cell line.

There is a drop in UPEC associated with Rab27b+ vesicles from 70% at 5h to 30% at 12h as shown in 2B. However, this is not supported by the data shown in Fig 2E. Please discuss this. Also, why only one early timepoint for Fig 2D and why not have similar time points for 2D and 2E?

In Fig 2G, how can you calculate %UPEC/total infected BEC for TritonX100 treated wells? Won’t Triton treatment break open ALL cells, and total infected BEC will be 0? This is a minor concern that will be resolved by clearly explaining how the calculations are done in this and other figures such as Fig 3H.

Include complemented hlyA KO CFT073 containing plasmid-borne hlyD to confirm that hlyD KO does not have polar effects. The complemented strain should be included in BEC and in vivo experiments.

Label x-axis in Fig 3C. Please define it in the figure legend as well. Not clear how you reached MOI of 50 from Fig 3C.

Show limit of detection with a dotted line in Fig 6.

Did you detect any CFUs in kidneys?

Fig 6C, right panel. What is the meaning of y-axis label here?

Reviewer #2: 1. While the use of magnetic bead labeled UPEC is a new an interesting approach, there are experiments lacking which are necessary to validate this system. I don’t see data in this manuscript demonstrating that labeling UPEC with magnetic beads does not alter how UPEC attaches to and invades BECs. Are recovered CFUs similar to what is observed with unlabeled UPEC? Do the magnetic bead UPEC express type 1 fimbriae and other virulence factors (like HylA) similarly to unlabeled UPEC? Also, for all of the magnetic pull down experiments, a “vehicle” control where magnetic beads are added to BECs (without UPEC) needs to be performed. This is important as a non-specific binding control to determine if the beads are taken up into vesicles by themselves (independent of UPEC) and if they alone can pulldown RAB27b fusiform vesicles for example. Also, when performing these types of experiments, it is usually good practice to perform blots for markers of other cellular compartments you do not expect the bacteria to be housed in to ensure specificity.

2. Maybe I am missing something, but I don’t understand why it is assumed there is a cytosolic intermediate for UPEC in BECs. The authors interpret the data in figure 2 to mean that UPEC escapes from RAB27b vesicles into the cytosol where it is recaptured into LC3A/B compartments. Couldn’t an alternate hypothesis be that the UPEC-containing RAB27b vesicles are trafficked and develop into LC3A/B compartments as part of the endocytic pathway? Even with evidence for a cytosolic stage (which I believe is in Figure 2G but it’s hard to tell because this figure is not called out where this is discussed in the results), how could one be certain that the bacteria individually transition from RAB27b to cytosol to LC3A/B compartments without a time-lapse experiment that tracks individual bacteria over time?

3. For lanes to be qualitatively compared in a Western blot they must be on the same blot (upper and lower panels of 3E cannot be compared with each other for example as the exposure time or amount loaded seems to be different). These blots need to be redone in this configuration if comparisons are to be made. Also, the authors overstate the observed differences in band intensities in figures 3E and 4G.

4. I am concerned about the observed CFU in the mouse bladder infection experiments in Figures 1 and 6. These are too low for and do not reflect is normally observed with UPEC infection of C57BL6 mice, especially for CFT073 (Figure 6). Given these abnormally low bladder bacteria titers, it is difficult to interpret the results of the mouse experiments. Perhaps the bladders are not being properly inoculated or there is a difference in the how the bacteria are being prepared for inoculation?

5. A critical control is missing for Figure 1D. Without the inclusion of a non-specific control (secondary antibody only), it is impossible to determine if the observed fluorescence is bacteria or autofluorescence which is common in bladder tissue. By just looking at the size, it seems as if these structures are quite a bit larger than the bacteria observed in the BECs in Figure 2A for example. Also, unless this was the only case and only field in which signal was detected the authors should provide additional representative images from other fields and other patient biopsies in the supplement to further support these findings.

Reviewer #3: Different methods were applied to very narrow time points, and the findings would be more convincing if all salient markers (Rab27b, LC3A/B) were measured at each timepoint to show how they are changing together—you could use a ratio of each marker normalized to a control. How many times were these pulldowns/blots conducted? Densitometry from several blots can be shown alongside a single blot. If bacteria are supposed to be in the cytosol vs vesicles at specific timepoints, show gentamicin sensitivity changing over time instead of providing a single snapshot. Data should be provided for 1) total CFU, 2) intracellular CFU, and 3) cytosolic vs vesicle-associated CFU so it is clear whether bacteria are invading and surviving normally.

Additional (or better) controls would also strengthen the conclusions. For example, uninfected or mock-infected cells are an important control that needs to be added for most of the experiments (see especially Fig 4). Throughout the manuscript, GAPDH is used as a control for western blots (as stated in the Fig 2 legend, although this antibody was not mentioned in the list of antibodies in the methods), but this antibody typically reacts with both mammalian and bacterial GAPDH. (note, the control should be stated clearly for the blot in Fig 3E.) Thus, it is not clear what is being assayed when, for example, magnetically labeled bacteria are used to pull down host cell components. Separate controls should be used to show both host cell protein levels and bacterial enrichment. The anti-E. coli antibody that seems to have been used for immunofluorescence microscopy might be a suitable alternative.

There needs to be more work to place this study into the body of knowledge for UPEC-bladder urothelium interactions. For example, quiescent intracellular reservoirs (QIRs) have been described in chronic/recurrent UTI mouse models. Similarly, the cell line used here is derived from a carcinoma and is known to have altered autophagic response. These cells also lack the fusiform vesicle physiology of umbrella cells, and they do not assemble urothelial plaques. It is not clear if the mechanism translates to intact, polarized, uroplakin crystal-coated bladder cells. Likewise, use of pan-inhibitors will be tremendously disruptive of normal cellular physiology, including the pathways and markers in this current study. Thus, claims that specific pathways are being targeted by bacteria are overstated. This does not negate the experiments conducted, but it needs to be acknowledged.

HlyA is not produced by the majority of UPECs (e.g., 31-48% PMID: 26299820, 20% PMID: 28330863) and, if anything, seems to be more correlated with pyelonephritis. Indeed, the prevalence of hlyA and its role in recurrence is overstated in the introduction: ref 19 is a lung study; ref 20 is PCR positivity, not expression, and does not demonstrate a significant increase in hlyA in patients with recurrent UTI. Thus, this proposed mechanism, if true, is restricted to a narrow subset of UPECs. It would be useful to show mouse bladder persistence for a panel of wild-type UPEC isolates as well as the 8-week persistence data that is mentioned in the first paragraph of the results but not shown. Do hlyA-negative UPEC isolates display similar persistence in the mouse model used here? This should be factored into the vaccine discussion as well; how likely is a secreted hemolysin that seems to exert its most potent action intracellularly to be an effective vaccine antigen? Finally, it is not clear how a pore-forming toxin is interfering with recruitment of V-ATPase to lysosomes. It would be helpful to propose a model that could be tested.

Please add size bars to all micrographs. In addition, bacteria are nearby the host cell vesicular membrane markers, but not colocalizing (this would mean the markers are superimposed, not adjacent). The figure legends state that confocal microscopy was used; z-stacks should therefore be collected to show that bacteria are inside vesicles. How many fields were examined for each set of experiments?

Please add discussion about possible toxic effects of microtubule stabilization on the bladder, particularly since this is being proposed as safer than antibiotic treatment. In addition, the current work must be better placed into the context of HlyA and bladder persistence work done by others in the UTI field (PMID 25675528). It would be helpful to discuss how Rab27b+ positivity, which is used for apical targeting of fusiform vesicles (e.g., PMID 27009205), is tied to bacterial internalization.

**Part III – Minor Issues: Editorial and Data Presentation Modifications**

Reviewer #1: 1. On page 6, please replace capacity with ability in the following sentence: “Thus, the capacity of UPEC CI5 to persist within human BECs in vitro appeared to mimic their capacity to persist within the superficial bladder epithelium in mice.”

2. Please rewrite the following sentence, it is very difficult to understand.

Page 6- “These findings suggest that evading vesicles that encase UPEC can benefit the bacteria, as they assist in avoiding elimination by BECs.”

3. Page 7: “Bacterial growth was minimal in siRNA-transfected BECs, but was significantly higher in BECs where autophagosomes were deficient and many more cytosolic bacteria were present (Fig 2H).” This sentence is difficult to understand. Minimal in reference to what? Which siRNA transfected BECs show minimal bacterial growth? Also “many more” is vague, please replace it with by “X-fold higher or lower”.

4. “then at a time when we expect UPEC to be cytosolic, solubilize BECs to release intracellular BECs and then isolate them using a magnet.” Correct this sentence.

5. Rephrase the following sentence. It is difficult to understand: “To prove that these isolated UPEC are not encased in any vesicle and are truly cytosolic, we exposed them to gentamicin, and expected them to be susceptible to gentamicin because they are not membrane-bound.”

6. Page 8: “we labeled the WT CFT073 and ΔhlyD mutant strains with magnetic beads and exposed them to BECs at an MOI of 20, for 2 h.” Exposing bacteria to BECs sounds counterintuitive. Rephrase this.

Reviewer #2: 1. Scale bars are missing from all micrographs.

2. Please provide a reference for the selective disruption of autophagosome membranes by Triton X-100

3. Line numbers should be added

4. Please explain more clearly what ATG5 is and why exactly knocking it down would reduce the number of autophagosomes – saying it is an essential component is not sufficient.

5. For the experiment in Figure 4H I think it would be more appropriate to pharmacologically inhibit the V-ATPase after infection has been initiated then to perform siRNA knockdown before the infection is started.

Reviewer #3: Fig 1B/C: bacteria that persist in the bladder could also be resistant to neutrophils, perhaps due to biofilm formation, fast growth, or toxins that directly affect neutrophils. Indeed, HlyA has been previously hypothesized to suppress inflammatory responses (PMID 22264513, which was cited in the discussion but not addressed in this context).

Fig 1E: LC3A/B should appear as a doublet, but only one band is shown in the very closely cropped image (the lower band might be barely visible). The ratio between these is useful to distinguish autophagocytic response. It would be helpful to show the full blot as a supplemental figure. Also, Fig 1E describes experiments out to 5 days p.i.; is it surprising that bacterial numbers in this cell culture model remained steady after 24h? Were intracellular bacterial communities ever observed?

The y-axis for several figures (e.g., Fig2B and C, Fig 3F and H) is confusing. It seems that the denominator is “total UPEC” but the axes show “/Total infected BECs.” If it is indeed total UPEC, how is that being quantified? Please clarify.

Figs 3A, B, C: please describe how hemolysis and cytotoxicity were assayed. Both techniques are absent from the methods section.

Fig 4D: Please comment on the apparent increase in LAMP1 at 48h.

Fig 4G: what is this time point? This is especially important since Fig 3 suggests there are different levels of wt vs mutant bacteria by 4 hpi.

Fig 6: methods need to be provided detailing the dosages and timing of paclitaxel administration.

Methods:

Please provide a reference for UPEC strain CI5 or provide further description of its origin.

As far as I am aware, Dr. Mobley is still at the University of Michigan, not Washington.

Was bacterial viability measured after magnetic labeling? What percentage of bacteria were labeled by the beads?

Please note inconsistent spelling of “gentamicin” throughout.

PLOS authors have the option to publish the peer review history of their article (what does this mean?). If published, this will include your full peer review and any attached files.

Reviewer #1: No

Reviewer #2: No

Reviewer #3: No
---

## [Decision Letter · Decision Letter 1]

4 Mar 2023

Dear Professor Choi,

Thank you very much for submitting your manuscript "α-Hemolysin Promotes Uropathogenic E. coli Persistence in Bladder Epithelial Cells Via Abrogating Bacteria-Harboring Lysosome Acidification" for consideration at PLOS Pathogens. As with all papers reviewed by the journal, your manuscript was reviewed by members of the editorial board and by several independent reviewers. The reviewers appreciated the attention to an important topic. Based on the reviews, we are likely to accept this manuscript for publication, providing that you modify the manuscript according to the review recommendations.

Authors should modify the text to improve clarity, and to account for potential limitations and alternative explanations for their results noted by Reviewers.

Sincerely,

Sargurunathan Subashchandrabose

Guest Editor

PLOS Pathogens

Brian Coombes

%CORR_ED_EDITOR_ROLE%

PLOS Pathogens

Kasturi Haldar

Editor-in-Chief

PLOS Pathogens

orcid.org/0000-0001-5065-158X

Michael Malim

Editor-in-Chief

PLOS Pathogens

orcid.org/0000-0002-7699-2064

Authors should modify the text to improve clarity, and account for potential limitations and alternative explanations for their results noted by Reviewers.

Reviewer Comments (if any, and for reference):

Reviewer's Responses to Questions

**Part I - Summary**

Reviewer #1: This is a revised version of the manuscript detailing the molecular stages during the bladder epithelial infection by UPEC. The authors have carefully answered every concern raised by me and by other reviewers. However, I still have concerns

Reviewer #2: The authors have fully addressed my concerns.

Reviewer #4: This manuscript “�-Hemolysin promotes uropathogenic E. coli persistence in bladder epithelial cells via abrogating bacteria-harboring lysosome acidification” by Naskar et al, describes the intracellular lifecycle of UPEC. Specifically, they use cellular and molecular biology techniques to probe the vesicular compartments of the host cell to assess the location of UPEC. This study uses a variety of complimentary techniques to answer very specific questions about the stages of intracellular movement. They conclude that although initially encased in RAB27b+ vesicles, UPEC briefly escapes into the cytosol before recapture in the autophagosome. Here, UPEC can prevent acidification and maturation of the autolysosome by inhibiting host V-ATPase via microtubulin instability. From the bacterial perspective, they use UPEC strain CFT073 and an isogenic hemolysin mutant to state that the hemolysin toxin is critical for RAB27b+ vesicle escape and prevention autolysosome acidification. The ultimate conclusion is that hemolysin facilitates long-term persistence in the host bladder.

While several sections of this manuscript have been revised and/or improved by additional experiments, there still remain critical points highlighted during the initial review that were not adequately addressed.

In addition, the techniques are very robust and novel; however, there are still concerns that the conclusions drawn by the authors are not fully supported by the data presented, and several key experiments are lacking appropriate cellular controls. There must be more textual clarity and mention of the experimental limitations present in this study.

**Part II – Major Issues: Key Experiments Required for Acceptance**

Reviewer #1: The authors mention that Figures 1A and 1C represent three and two experiments, respectively and each dot represents a single mouse. Did authors include 5 mice each in three and two replicates for Fig 1A and C? This should be clearly stated in Figure legend.

The authors also say "we have modified the bar graph in Figure 1A with the median as the central tendency and SD."--This change is not reflected in the figure legend. This needs to be corrected. Also, standard deviation should be removed and median should be shown as a line and not as a bar.

I agree with the authors that "The Mann-Whitney U test is used to compare differences between two independent groups when the dependent variable is either ordinal or continuous, but not normally distributed." Since the mouse CFU data are not normal, they should use Kruskal-Wallis test.

Reviewer #2: NA

Reviewer #4: Major Comments:

1. Results section “Trafficking of intracellular UPEC from RAB27b vesicles into LC3A/B+ compartments is proceeded by a brief cytosolic phase”

It is unclear if autophagic and RAB27b+ vesicles co-localize. There is no microscopy image that shows both markers simultaneously and Figure 2D does not preclude that both markers are present on the same vesicle. Since autophagy is a known protein recycling pathway, it is plausible that LC3A/B+ compartments encompass damaged (via hemolysin) RAB27b+ vesicles; therefore, there would be no true cytosolic stage for the bacteria. Because of this, the subsequent knock down of ATG5 only proves that RAB27b+ vesicles have been degraded, but not that a cytosolic stage exists when autophagy is functional. A differential centrifugation prep would be more conclusive to test for viable bacteria in the cytosolic fraction. This was a missed opportunity that could have been used in conjunction with targeted enrichment for vesicle-associated UPEC.

The field uses 5637 cells; however, it must be mentioned somewhere (in addition to the statements in lines 408-411) the considerations of these cells being of cancerous origin. It is known that many cellular pathways (including autophagy) function differently in cancer cells.

2. Results section “HlyA facilitates the escape of UPEC from RAB27b+ vesicles”

To support this claim, authors must include the genetic complementation of CFT in these specific, keystone experiments. Ideally, there would be similar studies in a HlyA- UPEC strain as a control to demonstrate that these phenomena are truly hemolysin-mediated.

The cell monolayer would be partially permeable or “leaky” (even though sublytic) with many of these assays due to hemolysin insertion into the plasma membrane. At what point does it become gentamicin sensitive in this process? Similarly, the RAB27b+ vesicle would also go through various stages of degradation via hemolysin.

Why the switch from MOI 50 to 20?

The data in Figure 3G are highly variable and are not truly indicative of expulsion (rather than IBC formation followed by cell death) without more data. The clusters of bacteria are visible in most of the microscopy images. QIRs have been described as single bacteria in a vesicle. IBC are large clusters that grow in the cytoplasm until the cell bursts. How do these fit into the paradigm? Although there is brief mention, more explanation is needed.

3. Results section “Inhibition of V-ATPase recruitment by HlyA” and “Destabilization of the cytoskeletal structure in bacteria-infected BECs reduces intracellular bacterial killing”

The narrative mentions expression of hemolysin while the UPEC are intracellular; however, the experimental design does not test this directly. This is critical due to the verbiage in this section and in the discussion. The vesicle isolation preps could have been blotted for HlyA and/or a luminescent reporter containing the hlyA promoter could have been used.

We can’t be sure that the hlyD mutant doesn’t have altered regulation of other virulence factors preventing acidification. These experiments must be performed with the complemented mutant to support these statements.

The wording makes it seem as if HlyA has a direct interaction with V-ATPase at this particular stage of the intracellular lifecycle. Could it not be the hemolysin inserted into the plasma membrane that is destabilizing the cytoskeleton and thus V-ATPase recruitment? The findings in Fig. 5A demonstrate that the cell architecture has been compromised. It could be from the effects of hemolysin secreted extracellularly leading to microtubule dysregulation inside the cell.

With many of these experiments, plasma membrane was selectively disrupted when it could have been isolated and/or stained as a negative internal control. Couldn’t the gentamicin-sensitive UPEC be associated with the membrane? A co-IP for plasma membrane associated protein (clathrin/calveolin/dynamin) could address this.

4. Based on the fact that only 30-50% (as stated in the introduction) of UPEC are hlyA+, some of the statements about future therapeutics are not fully supported by these data.

**Part III – Minor Issues: Editorial and Data Presentation Modifications**

Reviewer #1: In general, there are a lot of changes in figures although they are not always reflected in the figure legend. Atleast they are highlighted in yellow.

Include limit of detection in ALL figures showing CFU/ml data-Fig 1A, 1C, Fig 6C, etc. Explain in methods how the LOD was determined.

Reviewer #2: The authors have fully addressed my concerns

Reviewer #4: Minor Comments:

• I assume CI5 is hlyA+, but it is not explicitly stated

• Add Fig S1 to panel Fig 1A

• Line 103-104: replace with “neutrophil recruitment had subsided”

• rUTI patient samples during an episode of UTI or during “dormancy”?

• What is the genotype (hlyA+ or -) of rUTI strains shown here?

• Were gentamicin concentrations tested on CI5 to determine sensitivity?

• Why include J96 (fig. 3A) if there is no mention in the text?

• Line 114: What is this number as a percent of total UPEC infected? (only 0.01% of UPEC enters the cell at MOI 100)

• Triton-X should have been an included positive control in cytotoxicity assays

• Clusters of UPEC in autolysosomes (as seen in microscopy images) would skew these CFU data

• Figure 6B can’t be 0 CFU unless the entire organ homogenate was plated, which is not the case according to the methods

• Gentamicin misspelled in figures

• Does Fig. S20 indicate that there is only n=1 for WB experiments?

PLOS authors have the option to publish the peer review history of their article (what does this mean?). If published, this will include your full peer review and any attached files.

Reviewer #1: No

Reviewer #2: No

Reviewer #4: No

Figure Files:

Data Requirements:

Reproducibility:

References:

---

## [Editor Report · Decision Letter 2]

25 Apr 2023

Dear Professor Choi,

We are pleased to inform you that your manuscript 'α-Hemolysin Promotes Uropathogenic E. coli Persistence in Bladder Epithelial Cells Via Abrogating Bacteria-Harboring Lysosome Acidification' has been provisionally accepted for publication in PLOS Pathogens.

Best regards,

Sargurunathan Subashchandrabose

Guest Editor

PLOS Pathogens

Brian Coombes

%CORR_ED_EDITOR_ROLE%

PLOS Pathogens

Kasturi Haldar

Editor-in-Chief

PLOS Pathogens

orcid.org/0000-0001-5065-158X

Michael Malim

Editor-in-Chief

PLOS Pathogens

orcid.org/0000-0002-7699-2064
---

## [Editor Report · Acceptance letter]

9 May 2023

Dear Professor Choi,

We are delighted to inform you that your manuscript, "α-Hemolysin Promotes Uropathogenic *E. coli* Persistence in Bladder Epithelial Cells *Via* Abrogating Bacteria-Harboring Lysosome Acidification," has been formally accepted for publication in PLOS Pathogens.

Best regards,

Kasturi Haldar

Editor-in-Chief

PLOS Pathogens

orcid.org/0000-0001-5065-158X

Michael Malim

Editor-in-Chief

PLOS Pathogens

orcid.org/0000-0002-7699-2064